

# ENSO-ASC 1.0.0: ENSO Deep Learning Forecast Model with a Multivariate Air-Sea Coupler

Bin Mu[1*], Bo Qin[1*], and Shijin Yuan[1]

[1]School of Software engineering, Tongji University, Shanghai, 201804, China

[*]These authors contributed equally

*Correspondence to*: Shijin Yuan (yuanshijin2003@163.com)

**Abstract.** ENSO is an extremely complicated ocean-atmosphere coupling event, the development and decay of which are usually modulated by the energy interactions between multiple physical variables. In this paper, we design a multivariate air-sea coupler (ASC) based on graph using features of multiple physical variables. On the basis of the coupler, an ENSO deep

learning forecast model (named ENSO-ASC) is proposed, whose structure is adapted to the characteristics of the ENSO dynamics, including the encoder/decoder for capturing/restoring the multi-scale spatial-temporal correlations, and two attention components for grasping the different air-sea coupling strength on different start calendar months and varied contributions of physical variables in ENSO amplitudes. In addition, two datasets at different resolutions are used to train the model. We firstly tune the model performance to optimal and compare it with the other state-of-the-art ENSO deep learning

forecast models. Then, we evaluate the ENSO forecast skill from the contributions of different predictors, the effective lead time with the different start calendar months, and the forecast spatial uncertainties, further analyze the underlying ENSO mechanisms. Finally, we make ENSO predictions over the validation period from 2014 to 2020. Experiment results demonstrate that ENSO-ASC outperforms the other models. Sea surface temperature (SST) and zonal wind are two crucial predictors. The correlation skill of Niño3.4 index is over 0.78/0.65/0.5 within the lead time of 6/12/18 months. From two

heat map analyses, we also discover the common challenges in ENSO predictability, such as the forecasting skills declining faster when making forecasts through June-July-August and the forecast errors more likely showing up in the western and central tropical Pacific Ocean in longer-term forecasts. ENSO-ASC can simulate ENSO with different strengths, and the forecasted SST and wind patterns reflect obvious Bjerknes positive feedback mechanism. These results indicate the effectiveness and superiority of our model with the multivariate air-sea coupler in predicting sophisticated ENSO and

analysing the underlying dynamic mechanisms.

## 1 Introduction

El Niño-Southern Oscillation (ENSO) can induce the global climate extremes and ecosystem impacts (Zhang et al., 2016), which is the dominant source of interannual climate change. The El Niño is the ocean phenomena of ENSO, and is usually considered as the large-scale positive sea surface temperature (SST) anomalies in the tropical Pacific Ocean. Niño3 (Niño4)



index is the common indicator for ENSO research to measure the cold tongue (warm pool) variabilities, which is the averaged SST anomalies covering the Niño3 (Niño4) region (See Fig. 1). Besides these two indicators, the ONI (oceanic Niño index, 3 month running mean of SST anomalies in the Niño3.4 region) has become the de-facto standard to identify the occurrence of El Niño/La Niña event: If the ONIs of 5 consecutive months are over 0.5 ℃ (below -0.5 ℃), El Niño (La Niña) occurs.

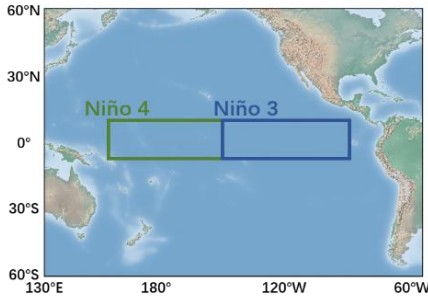


**Figure 1: Most concerned regions in ENSO events. The blue rectangle covers the Niño3 region (N5°-S5°, W150°-W90°), and the green rectangle covers the Niño4 region (N5°-S5°, E160°-W150°).**

Conventional forecast approaches majorly rely on numerical climate models. However, it is worth noting that the model biases of traditional approach have always been a problem for ENSO prediction (Xue et al., 2013). In addition, many other
factors also limit the ENSO predictability such as natural decadal variations in ENSO amplitude. For example, predictability tends to be higher when the ENSO cycle is strong than when it is weak (Barnston et al., 2012; Balmaseda et al., 1995; McPhaden, 2012). Recently, due to deluges of multi-source real-world geoscience data starting to accumulate, e.g. remote sensing and buoy observation, meteorological researchers are inspired to build more accurate and convenient data-driven models at a low computational cost (Rolnick et al., 2019), which leads to a wave of formulating ENSO forecast with deep
learning techniques, producing more skilful ENSO predictions (Ham et al., 2019).

In the field of deep learning, ENSO prediction is usually regarded as forecasting the future evolution tendency of SST and related Niño indexes directly, subsequently analyzing the associated sophisticated mechanisms and measuring the intrinsic characteristics such as intensity and duration. Therefore, the simplest but most practical forecast approaches can be divided into two categories: Niño index forecast and SST pattern forecast.

As for Niño index forecast, many favourable neural networks have made accurate predictions in 6, 9 and 12 months ahead. For instance, ensemble QESN (McDermott and Wikle, 2017), BAST-RNN (McDermott and Wikle, 2019) and LSTM (long short-term memory) (Broni-Bedaiko et al., 2019) are representative works. These studies demonstrate that the deep learning can well capture the nonlinear characteristics of non-stationary time series and attain outstanding regression on Niño index.

Notwithstanding the successful attempts on the Niño index regression, there still exist many pitfalls in measuring ENSO forecasts using only one single scalar, which may lead to the blind pursuit of the accuracy of a certain indicator while seriously hampering the grasp of underlying physical mechanism. Therefore, many studies are suggestive of exploiting





spatial-temporal dependencies and predicting the evolution of SST pattern. Ham et al. (2019) applies transfer learning (Yosinski et al., 2014) on historical simulations from CMIP5 (Coupled Model Intercomparison Project phase5, Bellenger et al., 2014) and reanalysis data with a CNN model to predict ENSO events, resulting in a robust and long-term forecast for up to 1.5 years, which outperforms the current numerical predictions. Mu et al. (2019) and He et al. (2019) both build a ConvLSTM (Shi et al., 2015) model to capture the spatial-temporal dependencies of ENSO SST patterns over multiple time horizons and obtain better predictions. Zheng et al. (2020) constructs a purely satellite data-driven deep learning to forecast the evolutions of tropical instability wave, which is closely related to ENSO phenomena, and obtain accurate and efficient forecasts. These deep learning models tend to simulate the behaviours of numerical climate models, the inputs of which are historical geoscience data and the outputs of which are the forecasted SST pattern.

The reason for the great progress in these works is no accident. On the one hand, the deep learning model has much more complex structures and can mine the complicated features hidden in the samples more effectively, which allows them to be substantially more expressive with blending the non-stationarity in temporal and the remote correlation in spatial. On the other hand, it is very convenient to migrate deep learning computer vision technologies to ENSO forecast due to the nature analogy between the format of image/video frame data and meteorological time-series grid data, which offers promise for extracting spatial-temporal mechanisms of ENSO via advanced deep learning technologies relatively easily. Therefore, the data-driven deep learning can be an alternative to traditional numerical models and a powerful tool for the ENSO forecasting.

However, there are still some obstacles in the deep learning modelling process for ENSO forecast. Very often, most existing models are confined to limited or even single input predictor, such as only use historical SST (and wind) data as the model input. Meanwhile, the climate deep learning models are rarely adaptively customized to the specific physical mechanisms of ENSO. These situations lead to poor interpretability and low confidence of ENSO-related deep learning models. ENSO is an extremely complicated ocean-atmosphere coupling event, the development and decay phases are closely associated with some crucial dynamic mechanisms and Walker circulation (Bayr et al., 2020), whose status have great impacts. Walker circulation is usually modulated by multi-physical variables (such as SST, wind, precipitation, etc.), and there are always coupled interactions between different variables. More specifically, the varieties in Walker circulation have strong temporal-lag effects on ENSO ("memory effects"); the position of the ascending branch is also a very important climatic condition for the occurrence of El Niño. Such priori ENSO knowledge has not been effectively used in deep learning model.

Therefore, in order to further improve the ENSO prediction skill, there is an essential principle that should be reflected in climate deep learning models: subjectively incorporating priori ENSO knowledge into the deep learning formalization and deriving hand-crafted features to make predictions.

In this paper, according to the important synergies of multiple variables in crucial ENSO dynamic mechanisms and Walker circulation, we select 6 indispensable variables (SST, u-wind, v-wind, rain, cloud, and vapor) that are induced from ENSO-related key processes to build a multivariate air-sea coupler (ASC) based on graph mathematically, which emphasizes


the energy exchange between multiple variables. We then leverage the coupler and build up the ENSO deep learning forecast model, named ENSO-ASC, with an encoder-coupler-decoder structure to extract the multi-scale spatial-temporal features of multiple physical variables. Two attention components are also proposed to grasp the different air-sea coupling strength on

different start calendar months and varied contributions of these variables. A loss function combining *MSE* and *MAE* is used to guide the model training precisely, *SSIM* (structural similarity) (Wang et al., 2004) and *PSNR* (peak signal to noise ratio) are used as metrics to evaluate the spatial consistency of the forecasted patterns.

Two datasets are applied for model training to ensure that the system forecast errors are fully corrected after the higher quality dataset. We first train the ENSO-ASC on the numerous reanalysis dataset from 1850.1 to 2015.12 and subsequently

on the high-quality remote sensing dataset from 1997.12 to 2012.12 for fine-tuning, this procedure is also known as transfer learning. The validation period is from 2014.1 to 2020.8 in remote sensing dataset. The gap between fine-tuning set and validation set is used to remove the possible influence of oceanic memory (Ham et al., 2019).

It is the first time to design a multivariate air-sea coupler considering energy interactions. we evaluate the ENSO-ASC from three aspects: Firstly, we evaluate the model performance from the perspective of model structure, including the input

sequence length, the benefits of transfer learning, multivariate air-sea coupler and the attention components, and tune the model structure to optimal. Then, we analyze the ENSO forecast skill of the ENSO-ASC from the meteorological aspects, including the contributions of different input physical variables, the effectiveness of forecast lead time, the forecast skill changes with different start calendar months, and the forecast spatial uncertainties. Subsequently, we make the real-world ENSO simulations during the validation period by tracing the evolutions of multiple physical variables. From the experiment

results, ENSO-ASC performs better in both *SSIM* and *PSNR* of forecasted SST patterns, which effectively raises the upper limitation of ENSO forecast. The forecasted ENSO events are more consistent with real-world observations and the related Niño indexes have higher correlations with observations than traditional methods and current state-of-the-art deep learning models, which is over 0.78/0.65/0.5 within the lead time of 6/12/18 months. SST and zonal wind are two crucial predictors, which can be considered as the major triggers of ENSO. A temporal heat map analysis illustrates that the ENSO forecasting

skills decline faster when making forecasts through June-July-August, and a spatial heat map analysis shows that the forecast errors are more likely to show up over the central tropical Pacific Ocean in longer-term forecasts. At the meanwhile, in the validation period from 2014 to 2020, the multivariate air-sea coupler can capture the latent ENSO dynamical mechanisms and provide multivariate evolution simulations with a high degree of physical consistency: The positive SST anomalies first show up over the eastern equatorial Pacific with the westerly wind anomalies in the western and central tropical Pacific

Ocean (vice versa in the La Niña events), which induces Bjerknes positive feedback mechanism. It is worth noting that for the simulation of 2015/2016 super El Niño, ENSO-ASC captures its strong evolutions of SST anomalies over the northeast subtropical Pacific in the peak phase and successfully predicts its very-high-intensity and very-long-duration, while many dynamic or statistical models fail. At the same time, ENSO-ASC can also reduce false alarm rate such as in 2014. From the mathematical expression, the multivariate air-sea coupler captures the spatial-temporal multi-scale oscillations of Walker

circulation and handle the ocean-atmosphere energy exchange simultaneously, which tries to avoid the interval flux



exchange in geoscience fluid programming of traditional numerical climate models. In conclusion, the graph-based multivariate air-sea coupler exhibits not only effectiveness and superiority to predict sophisticated climate phenomena, but also is a promising tool for exploiting the underlying dynamic mechanisms in the future.

The remainder of this paper is organized as follows. Section 2 introduces the proposed multivariate air-sea coupler.
Section 3 describes the ENSO deep learning forecast model with the coupler (ENSO-ASC) in detail. Section 4 illustrates the datasets, experiment schemas and result analyses. Finally, Section 5 offers further discussions and summarizes the paper.

## 2 Multivariate Air-Sea Coupler based on Graph

ENSO is the most dominant phenomenon of air-sea coupling over the equatorial Pacific, and many complex dynamical mechanisms modulate the ENSO amplitudes. Bjerknes positive feedback (Bjerknes, 1969) is one of the most significant
effects, the processes of which are highly related to the status of Walker circulation. There are energy interactions between the multiple physical variables influenced by Walker circulation every moment, and the ENSO-related SST varieties are greatly affected by such air-sea coupling (Gao and Zhang, 2017; Lau et al., 1989; Lau et al., 1996).

Many atmospheric and oceanic anomalies can be known as triggers of ENSO events, which establishes the Bjerknes positive feedback. The warming SST anomalies propagate to the central and eastern equatorial Pacific gradually. As SST
gradually rises, it is virtually impossible for the equatorial Pacific to enters a never-ending warm state. Therefore, some negative feedback will cause turnabouts from warm phases to cold phases (Wang et al., 2017). These negative feedback mechanisms all emphasize air-sea interactions. For example, westerly wind anomalies in the central tropical Pacific Ocean induce the upwelling Rossby and downwelling Kevin oceanic waves, both of which propagate and reflect on the continental boundary, then tend to push the warm pool back to its original position in the western Pacific. From the perspective of ENSO
life-cycle, atmospheric/oceanic variables play crucial roles together.

At the meanwhile, during the development and decay phase of ENSO, there also exist nonlinear interactions between atmospheric and oceanic variables. Wind anomalies are the most obvious and direct response of ENSO-driven large-scale oceanic varieties, and they will change the ocean-atmosphere heat transmission (Cheng et al., 2019). Once the ocean status changes, the thermal energy contained in the sea will escalate or lose into the air, hindering or promoting the precipitation
and surface humidity over the equatorial Pacific. These changes also give feedback on the ENSO.

Meteorological researchers have already identified the key physical processes in ENSO in recent years. If such knowledge can be incorporated into ENSO deep learning forecast modeling subjectively, breaking away from the current limitation of using single predictors, the accuracy of ENSO prediction will promise breakthroughs. In this paper, we choose 6 different ENSO-related indispensable variables from two different multivariate datasets as shown in Table 1, which have
strong correlations within the evolution of ENSO events according to Bjerknes positive feedback and other dynamical processes. Furthermore, in order to comprehensively represent the coupled interactions, a multivariate air-sea coupler $coupler(G)$ is designed to simulate their coupling synergies with an undirected graph $G = (V, A)$ as shown in Fig. 2, where



$V = (f_{v_1}, f_{v_2}, \dots, f_{v_N})$ represents the vertices of graph and $f_{v_i}$ is the feature of physical variables $v_i$ ($i = 1,2, \dots, N$). $A \in R^{N \times N}$ is the pre-designed adjacency matrix, where $A_{i,j} = 1$ ($A_{i,j} = 0$) represents there existing (not existing) energy interactions

between the connected variables $v_i$ and $v_j$. The variables exchange energy simultaneously every moment, and the directions of edges in this graph can be neglected because the energy interactions are two-way (transfer and feedback).

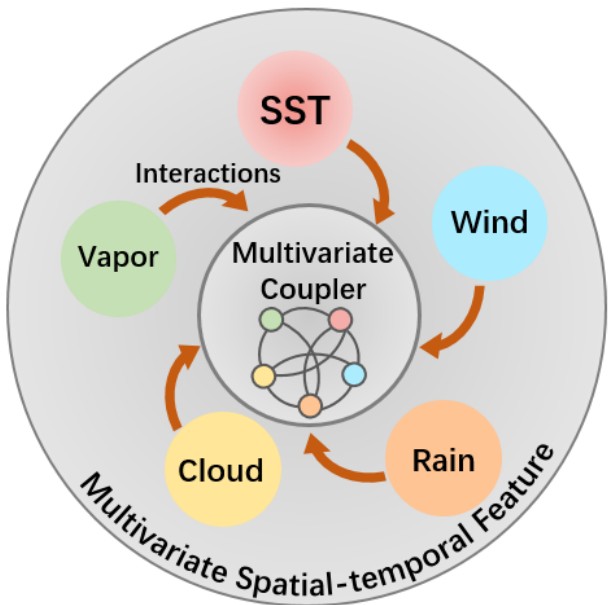

**Figure 2: Our proposed multivariate air-sea coupler, which utilizes the spatial-temporal features of multiple physical variables to simulate the energy exchanging simultaneously.**

Here, $V$ in $G$ is not the physical variable on a single grid point, but the features of the variable on the entire pattern. The reason lies in that: On the one hand, the coupler will pay more attention on the global/local spatial-temporal correlations in the variable fields of ENSO rather than the variations on an isolated grid. On the other hand, the coupler will provide a higher computational efficiency and consume a lower calculation resource for ENSO forecast. The finer and improved couplers, such as designing individual graph for smaller-scale regions and even a single grid, are future considerations.

**3 ENSO-ASC: ENSO Deep Learning Forecast Model with the Multivariate Air-Sea Coupler**

Inspired by previous ENSO deep learning forecast model, we can define the ENSO forecast as a multivariate spatial-temporal sequence forecast problem as illustrated in Eq. (1),

$$\hat{s} = \mathcal{F}_\theta(s^{scm}), \{sst, uwind, vwind, rain, cloud, vapor\} \subseteq s \qquad (1)$$

where $s^{scm} \in R^{N \times M}$ is $N$ multivariate observations in historical $M$ months ($N = 6$), $\hat{s} \in R^{N \times H}$ is the prediction result for

future $H$ months ($H$ can be also treated as forecast lead time). $scm \in \{Jan, Feb, \dots, Dec\}$ (start calendar month) represents the last month in the input series $s_{scm}$. $\mathcal{F}_\theta$ represents the forecast system ($\theta$ is the trainable parameters in the system).





In order to combine the multivariate coupler, we break down the conventional formulization and redefine the multivariate ENSO forecast model as the encoder-coupler-decoder structure shown as Eq. (3) to Eq. (5),

$$\text{encoder: } f_{v_i} = encoder_i(s_i^{scm}) \tag{2}$$

$$\text{coupler: } f_c = coupler(G) = coupler((f_{v_1}||f_{v_2}|| \dots ||f_{v_N}), A) \tag{3}$$

$$\text{decoder: } \hat{s}_i = decoder_i(f_{v_i}||f_{c_i}) \tag{4}$$

where the $s_i^{scm}(i = 1, 2, \dots, N)$ represents the individual physical data and $f_{v_i}$ represents the corresponding extracted features by respective encoders. The $coupler(\cdot)$ simulates the latent multivariate interactions on the physical features and the pre-designed interaction graph $G$, where the operator $||$ represents the concatenation of all features of physical variables. Then the respective decoders will restore the physical features end-to-end by the coupled multivariate features $f_{c_i}$ and original physical features $f_{v_i}$, the concatenation of which can be regarded as skip-layer connections. These connections can propagate the low level feature to high levels of the model directly, preserving the raw information and accelerate feature transfers to some extent. The sub-modules $encoder_i(\cdot)$, $coupler(\cdot)$, and $decoder_i(\cdot)$ form the ENSO-ASC together.

As mentioned before, the strength of the multivariate coupling and the influence of multivariate temporal memory in ENSO is changing with different input sequence length $M$ forecast start calendar month $scm$. In order to weaken such negative impacts on forecast, we design two self-supervised attention weights, $\alpha = (a_1, a_2, \dots, a_M)$ and $\beta = (b_1, b_2, \dots, b_N)$, in the encoder and coupler respectively to capture the dynamic time series non-stationarity and re-weight the multivariate contributions. The final forecast model can be written in shown in Eq. (5) to Eq. (7) as shown in Fig. 3, where ∘ represents the element-wise multiplication.

$$\text{encoder: } f_{v_i} = \alpha \circ encoder_i(s_i^{scm}) \tag{2}$$

$$\text{coupler: } f_c = coupler(\beta \circ G) = coupler((b_1 f_{v_1}||b_2 f_{v_2}|| \dots ||b_N f_{v_N}), A) \tag{3}$$

$$\text{decoder: } \hat{s}_i = decoder_i(f_{v_i}||f_{c_i}) \tag{4}$$

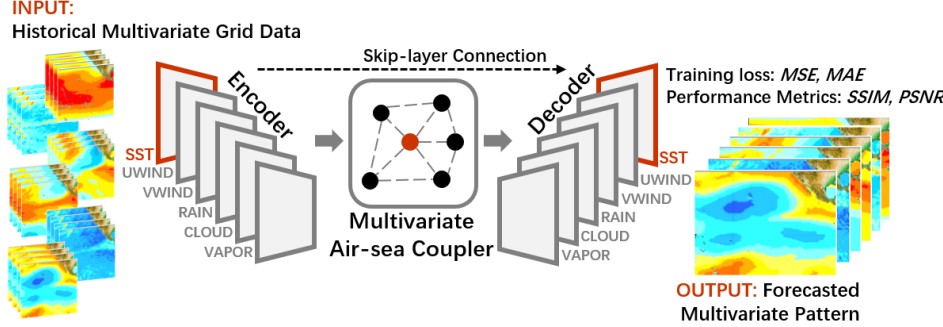

**Figure 3: The structure of ENSO-ASC. There are 6 encoders for the chosen variables to extract spatial-temporal information and a multivariate air-sea coupler to simulate interactions. After interactions, we design 6 decoders to restore the major variable SST and other variables.**

In addition, there are basically two forecast strategies for ENSO prediction: direct multi-step (DMS) and iterated multi-





step (IMS) (Chevillon, 2007). The former means predicting the future $H^{th}$-month multivariate pattern directly, and the latter means utilizing the forecast output result as the input for future iterated prediction. Figure 4 displays the differences between

DMS and IMS. In general, DMS is often unstable and more difficult to train for a deep learning model (Shi and Yeung, 2018). Therefore, we choose IMS to handle chaos data and provide more accuracy prediction, that is, outputting the next one-month's ($H = 1$) multivariate forecast result in the model, and then using this output as model input to continue forecasting the future evolutions. We also design a combined loss function to train our model and use two result metrics to evaluate the forecast results. The intentions and detailed implementations of every parts in the model are interpreted as the

following sections.

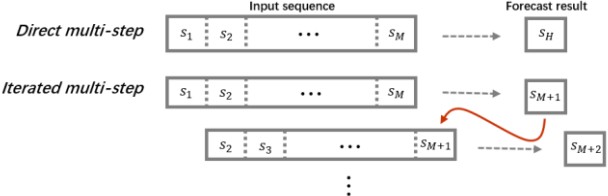

**Figure 4: Two common ways in sequence prediction such as ENSO forecast: Direct multi-step (DMS) and iterated multi-step (IMS).**

### 3.1 Encoder: Stacked ConvLSTM Layers for Extracting Spatial-Temporal Features

Considering the complex spatial-temporal characteristics of ENSO, the encoder is composed of three sub-structures: stacked ConvLSTM layers, 3D max-pool layers and the temporal attention weight.

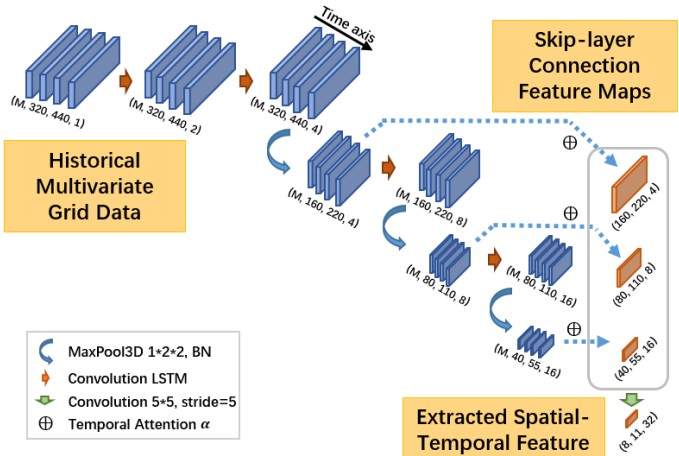

**Figure 5: A detailed structure for encoder: Stacked ConvLSTM encoder for extracting spatial-temporal features simultaneously. There is also temporal attention component for skip-layer connection in the grey box.**

The ENSO evolution has a strong correlation with historical atmospheric/oceanic memory (Zhang et al., 2019b). ENSO deep learning forecast model should be able to simultaneously extract the long-term spatial-temporal features from multivariate geoscience grid data and effectively mine the complicated nonlinearity hidden in the data. stacked ConvLSTM





layers are constructed as the skeleton of the encoder (See Fig. 5) for each chosen physical variable individually.

In order to capture the multi-scale spatial remote correlations at different resolutions in ENSO amplitudes, we take

advantage of a 3D max-pool layer between two ConvLSTM layers, the stride of which on time-axis is set to 1 to retain the

sequence length $M$. Considering these obtained multi-scale spatial features after every 3D max-pool layers, we design the

skip-layer connections shown in the grey box in Fig. 5. These layers pass and cascade the raw features of the same variable

from its encoder (lower levels) to its decoder (higher levels) directly (See Fig. 9) like dense connection (Huang et al., 2017).

Such structure can preserve more details at multi-scale spatial correlations and also solve the problem of gradient

disappearance. In addition, we design the encoders and decoders to be symmetric, ensuring that the feature maps in these

connections have the same shape.

Since we set $H = 1$ in the IMS forecast strategy, the feature maps on the encoder have a time axis, which the decoders

do not have. The memory effects on the forecast are mutative with input sequence length and start calendar month, if we

pass all time steps' feature maps in encoder to the decoder and the coupled module, it is too redundant and even causes over-

averaged forecast results, and the serious seasonal interference will also show up with different start calendar months.

Therefore, before skip-layer connections, we first determine the attention weights to dynamically fuse multiple time steps'

feature maps in encoder, which is called temporal attention weight $\alpha$ (as $\oplus$ symbols in Fig. 5). Meanwhile, this approach can

also dynamically extract strongly-correlated temporal memories and help improve the prediction skill.

After obtaining sequential feature maps $T = [t_1, t_2, \ldots, t_m], m = 1,2, \ldots, M$ from each 3D max-pool layer, we first flatten

every time step feature map $t_m \in R^{w \times h \times c}$ by the width $w$, height $h$ and channel $c$ as $t'_m \in R^{1 \times (w \times h \times c)}$, then connect them

together alone time axis as $T' = [t'_1, t'_2, \ldots, t'_m], m = 1,2, \ldots, M$, where $T' \in R^{M \times (w \times h \times c)}$. We apply Eq. (8) on $T'$ to

determine the self-supervised attentive weight $\alpha \in R^M$ for each time step's feature map $t_m$.

$$\alpha = softmax(W_{\alpha t} \, tanh(W_t T' + b_t) + b_{\alpha t}) \tag{8}$$

where $W_t \in R^{d_1 \times M}$ and $W_{\alpha t} \in R^{d_1}$ are transformation matrices, $d_1$ is a hyper parameter, $b_t \in R^{d_1}$ and $b_{\alpha t} \in R$ are biases.

Every dimension in $\alpha$ represents the contribution to the forecast of corresponding time step, so we use Eq. (9) to fuse the

original feature $T = [t_1, t_2, \ldots, t_m], m = 1,2, \ldots, M$.

$$\tilde{T} = h(\alpha, T) = \sum_{m=1}^{M}(\alpha_m \circ t_m) \tag{9}$$

where $\tilde{T}$ is the aggregated feature map for skip-layer connection, function $h(\cdot)$ represents the summary of element-wise

multiplication.

The feature map sizes are described in the Fig. 5 in detail. The sizes of ConvLSTM kernels are all $3 \times 3$ and the

channel sizes are $[2,4,8,16]$ during forward propagation, where the changes between adjacent layers are smooth and small.

The final output (with size of $16 \times 40 \times 55$) of the encoder is generated by a convolution layer of $32 \times 5 \times 5$ with stride 5

and output a feature map with size of $8 \times 11 \times 32$, which can reduce the noise derived by such the deep-layer structure.



## 3.2 Multivariate Air-Sea Coupler: Learning Multivariate Synergies via Graph Convolution

From the perspective of ENSO dynamics, the occurrences of ENSO are accompanied by energy interactions. Based on our formalization and chosen physical variables, we define the corresponding adjacency matrix $A \in R^{6 \times 6}$ and degree matrix $D \in R^{6 \times 6}$ (the diagonal matrix, the value on the diagonal indicates the number of other vertices connected to this vertices) as Eq. (10), which means all physical variables have coupled interactions with each other (full-connected).

$$A = \begin{bmatrix} 1 & \cdots & 1 \\ \vdots & \ddots & \vdots \\ 1 & \cdots & 1 \end{bmatrix}, D = \begin{bmatrix} 6 & \cdots & 0 \\ \vdots & \ddots & \vdots \\ 0 & \cdots & 6 \end{bmatrix} \tag{10}$$

In practical implementation mathematically, we use graph Laplacian matrix $L$ to normalize the energy flow of original adjacency matrix $A$ as Eq. (11). $L$ can be considered as the directions in which the excess unstable energy will propagate to other variables when the entire system is perturbed (such as external wind forcing).

$$L = I_n - D^{-\frac{1}{2}} A D^{-\frac{1}{2}} \tag{11}$$

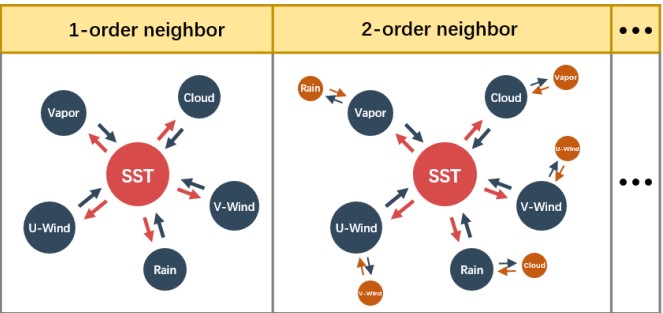

**Figure 6: Multivariate coupled interactions within K-order neighbours (take SST as the center).**

Meanwhile, the interactions between ENSO-related variables are cascaded. For example, the precipitation anomalies affect the wind anomalies, which in turn affects SST, as depicted in Fig. 6. According to the properties of Laplacian matrix, $L^k$ is employed to determine the cascaded interactions between $k$-order neighbors. So, if we consider $K$-order neighbors, the whole process is defined by Eq. (12),

$$P^{(l+1)} = \sigma(\sum_{k=1}^{K} \Theta^{(k)} L^k P^{(l)}) \tag{12}$$

where $\Theta^{(k)}$ represents the latent trainable multivariate interactions. $K$ represents the truncated order of neighbours concerned. $P^{(l)}$ represents the input features and $P^{(l+1)}$ represents the coupled features. Each row in both $P^{(l)}$ and $P^{(l+1)}$ represents the same variables. Activation function $\sigma$ increases the nonlinearity. Figure 7 illustrates the above process mathematically.





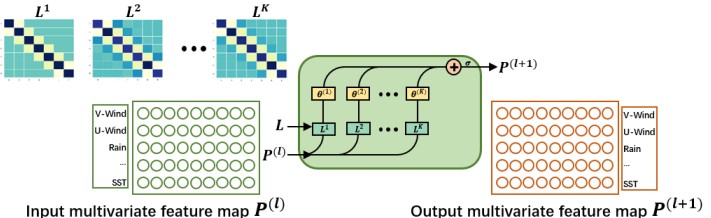

Input multivariate feature map $P^{(l)}$  Output multivariate feature map $P^{(l+1)}$

**Figure 7: Graph convolution layer. $L^k$ represents the interactions between $k$-order neighbours and each row in $p^{(l)}$ represents features of different physical variables.**

In deep learning, graph convolution network (GCN, Bruna et al., 2013) is applied to implement Eq. (12). Furthermore, we use Chebyshev polynomial $T_k(\tilde{L})$ to approximate the higher-order polynomial $[L^1, L^2, ..., L^k], k = 1,2, ..., K$ for accelerate calculation, where $\tilde{L} = 2L/\lambda_{max} - I_n$ scales $L$ within $[-1,1]$ for satisfying Chebyshev polynomial and $\lambda_{max}$ is the maximum

Eigen value of $L$ (Hammond et al., 2011; Defferrard et al., 2016). This approximation makes the graph convolution more promising by reducing the computational complexity from $\mathcal{O}(n^2)$ to $\mathcal{O}(K|\varepsilon|)$ ($|\varepsilon|$ is the edge count in the graph). Based on such neural structure, we construct the multivariate air-sea coupler (ASC) to learn synergies related to ENSO as Fig. 8.

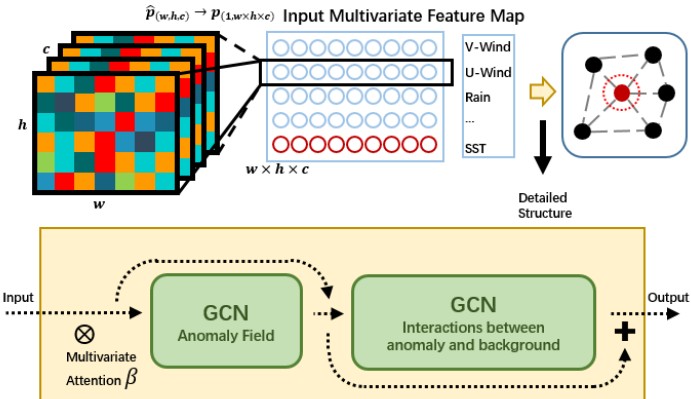

**Figure 8: A detailed structure for multivariate air-sea coupler between encoder and decoder: Pre-processes for input (upper row),**
**and dual-layer structure for simulating interactions between anomalous and background fields (lower row). There is also a multivariate attention component in the coupler.**

After obtaining spatial-temporal features from multivariate encoders respectively, we first flatten and cascade them as $P^{(l)}$ (See Fig. 8 (upper row)). As mentioned above, each row of $P^{(l)}$ represents different variables. $P^{(l)}$ is marked as multivariate feature map and acts as the input of the coupler. The coupler is designed as a dual-layer summation structure

(See Fig. 8 (lower row)), which is the residual learning to enhance the generalization ability of the network (He et al., 2016). The output of the multivariate coupler is determined by the weighted fusion of the outputs of these two layers as Chen et al. (2019).

Because variables contribute differently to the ENSO forecast especially in different start calendar months, we propose multivariate self-supervised attention weight for determining the weights for the input physical variables (as $\otimes$ symbols in





Fig. 8). Before $P^{(l)}$ passing into the multivariate coupler, the weight $\beta \in R^N$ for each variable is determined by Eq. (13).

$$\beta = softmax(W_{\beta p} \, tanh(W_p P^{(l)} + b_p) + b_{\beta p}) \tag{13}$$

where $W_p \in R^{d_2 \times N}$ and $W_{\beta p} \in R^{d_2}$ are transformation matrices, $d_2$ is a hyper parameter, $b_p \in R^{d_2}$ and $b_{\beta p} \in R$ are biases. Then we use Eq. (14) to calculate the modulated multi-physical variables feature map $P^{(l)}$, where $g(\cdot)$ represents the element-wise multiplication.

$$P^{(l)} = g(\beta, P^{(l)}) = \beta \odot P^{(l)} \tag{14}$$

In the multivariate air-sea coupler, the corresponding locations of physical variables on the input feature map and output feature map are fixed. For example, if we set the SST feature in the last row of the input multivariate feature map, the SST feature will be in the last row of the output multivariate feature map (See Fig. 7), which will be passed to the decoder later.

### 3.3 Decoder: End-to-end Learning to Restore the Forecasted Multivariate Patterns

ENSO evolution is considered as a hydrodynamic process. Meteorologists usually use linear methods (such as EOF or SVD method) to extract features, then analyze the potential characteristics and predict the future evolution of ENSO. In these methods, complex dynamical processes are usually simplified to facilitate calculations while unknown detailed processes are not revealed or even neglected, which leads to low prediction accuracy. Therefore, we use the end-to-end learning to restore the evolutions of multi-physical variables. The multi-scale spatial-temporal correlations should be also considered in this process, so the decoder consists of stacked transform-convolution layers and up-sampling layers.

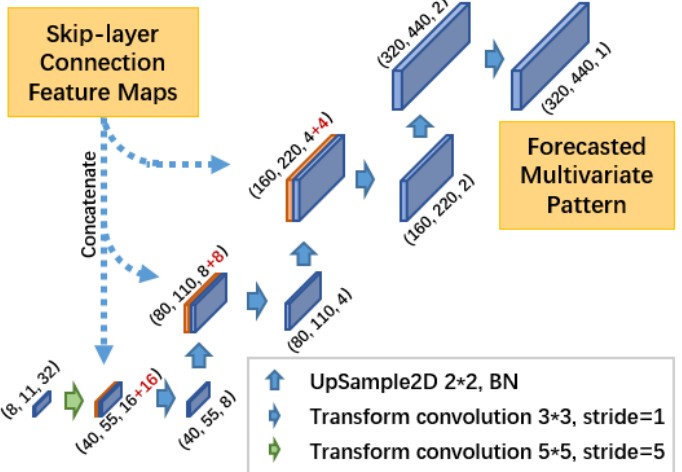

**Figure 9: A detailed structure for decoder, skip-layer connections (the red channel cascaded to the feature maps) from encoder for helping end-to-end learning to restore the forecasted patterns at different spatial scales.**

From the output feature map of the multivariate air-sea coupler, we pick up the corresponding row (taking SST as an example) as $P_{SST}$ (such as red row and red circle in Fig. 8), and reshape it into original shape $P'_{SST} \in R^{w \times h \times c}$. Then $P'_{SST}$ is gradually amplified and restored in the decoder with stacked transform-convolution layers and up-sampling layers (See Fig.



9). Skip-layer feature maps from encoder are cascaded with corresponding layers. The sizes of convolution kernels are all 3×3 which is the same with that in encoder, and the channel sizes are [16,8,4,2,1] to shrink the channel size gradually during forward propagation.

### 3.4 Loss Functions for model training

Our goal is to predict the evolutions of multiple physical variables (marked as $\hat{s}$) as accuracy as possible compared with the real-world observation $s$. Therefore, we combine two different measurements together as the loss function $\ell$ in Eq. (15) to ensure the spatial consistency of restored multivariate patterns grid by grid,

$$\begin{cases} MSE = \frac{1}{N\Omega}\sum_N \sum_\Omega (\hat{s}_{i,j} - s_{i,j})^2 \\ MAE = \frac{1}{N\Omega}\sum_N \sum_\Omega |\hat{s}_{i,j} - s_{i,j}| \\ \ell = MSE + MAE \end{cases}, (i,j) \in \Omega \tag{15}$$

where $N$ is the number of variables and $\Omega$ represents all grid points of every physical pattern, $(i,j)$ represent different latitude and longitude. $\ell$ is consist of the sum of $MSE$ and $MAE$, where $MSE$ is used to preserve the smoothness of the forecasted patterns and $MAE$ is used to retain the peak distribution of all grid points.

### 3.5 Metrics to evaluate the forecast results

According to the loss function, the calculation processes in Eq. (15) are mainly focused on the comparisons of single grid in fields. However, the detailed spatial distributions of every physical variable, such as the max value region of SST and the anomaly regions of wind, are more important in the ENSO forecast. Therefore, we especially use these two common quality metrics for the forecasted SST patterns to evaluate the ENSO forecast skill: $PSNR$, and $SSIM$ as Eq. (16) and Eq. (17),

$$PSNR = 10 \cdot log_{10}(\frac{MAX^2}{MSE}) \tag{16}$$

$$\begin{cases} luminance = \frac{2\mu_{\hat{s}}^2 \mu_s^2 + c_1}{\mu_{\hat{s}}^2 + \mu_s^2 + c_1} \\ contrast = \frac{2\sigma_{\hat{s}s} + c_2}{\sigma_{\hat{s}}^2 + \sigma_s^2 + c_2} \\ structure = \frac{\sigma_{\hat{s}s} + c_3}{\sigma_{\hat{s}}\sigma_s + c_3} \\ SSIM = luminance^a \cdot contrast^b \cdot structure^c \end{cases} \tag{17}$$

In $PSNR$, $MAX$ is the maximum of all grids. In ENSO-ASC, before the historical multivariate data input into the model, we first normalize them in the range [0,1] as Eq. (18). Therefore, $MAX$ is set to 1.

$$x^* = \frac{x - x_{min}}{x_{max} - x_{min}} \tag{18}$$

In $SSIM$, $\mu_{\hat{s}}$ ($\mu_s$) is the average for $\hat{s}$ ($s$), and $\sigma_{\hat{s}}$ ($\sigma_s$) is the corresponding standard deviations. $\sigma_{\hat{s}s}$ represents the covariance, $a = b = c = 1$ for fair measurement of every ingredient of $SSIM$, $c_1$, $c_2$, and $c_3$ are all with trivial values for preventing the denominator from being 0.


Besides these two metrics, the correlations between the calculated and the official Niño3.4 index will also be used to evaluate forecast skills.

## 4 Experiments Results and Analysis

### 4.1 Dataset Description

After the deep learning model structure is determined, the quality and quantity of training dataset affect the forecast performance decisively. As the improvement of observation ability, there are growing ways to provide multiple real-world observations, such as remote sensing satellite, buoy observation, which is more and more beneficial to build our ENSO forecast model. However, one of the biggest limitations in high-quality climate dataset is that the real-world observation period is too short to provide adequate samples. For example, extensive satellite observations may have started in 1980s, and

the number of El Niño that occurred ever since then is also small, which is easy to cause under-fitting of the deep learning network.

**Table 1: Multi-physical variables in the corresponding two datasets**

| NOAA/CIRES | | REMSS | |
|---|---|---|---|
| Variable | Description | Variable | Description |
| *SST* | Sea surface temperature | *SST* | Sea surface temperature |
| *PWAT* | Precipitation Water (Atmospheric Column) | *RAIN* | Rate of liquid water precipitation |
| *CWAT* | Cloud Water (Atmospheric Column) | *CLOUD* | Total cloud liquid water (Atmospheric Column) |
| *RH* | Surface relative humidity | *VAPOR* | Total gaseous water (Atmospheric Column) |
| *UWIND* | Surface zonal wind speed | *UWIND* | Zonal wind speed |
| *VWIND* | Surface meridional wind speed | *VWIND* | Meridional wind speed |

Note: The reanalysis dataset is provided by NOAA/CIRES, which is from 1850.1 to 2015.12 with 2 by 2 degree, and the remote sensing dataset is provided by REMSS, which is from 1997.12
to 2020.8 with 0.25 by 0.25 degree. In REMSS, UWIND and VWIND is collected from REMSS/CCMP (https://rda.ucar.edu/datasets/ds131.2/index.html), and other variables are collected from REMSS/TMI (1997-2012, http://www.remss.com/missions/tmi/) and REMSS/AMSR2 (2012-2020, http://www.remss.com/missions/amsr/). We try to choose physical variables in NOAA/CIRES with the same meaning with that in REMSS, such as CWAT and CLOUD, RH and VAPOR. Limited by these two datasets, some variables can only find the closest match though they describe the different characteristics in ocean-atmosphere cycle, such as PWAT and RAIN.

To greatly increase the quantity of training data, we utilize the transfer learning technique to train our model with long-term climate reanalysis data and high-resolution remote sensing data together. These datasets both provide multivariate





global observations. The reanalysis data is supported by NOAA/CIRES (https://rda.ucar.edu/datasets/ds131.2/index.html),
        which is a 6-hourly multivariate global climate dataset from 1850.1 to 2015.12 with 2° resolution. The remote sensing data
        is obtained from Remote Sensing System (REMSS, http://www.remss.com/), which is a daily multivariate global climate
        dataset from 1997.12 to 2020.8, and the resolution is much higher than reanalysis data with 0.25°. According to our chosen
        physical variables, we obtain the corresponding sub-datasets, and all the variables are preprocessed monthly-averaged. The
detailed dataset descriptions are shown in Table 1. Note that we try to choose physical variables in NOAA/CIRES with the
        same meaning with that in REMSS, such as CWAT and CLOUD, RH and VAPOR. Note that some variables can only find
        the closest match in these two datasets though they describe the slightly different characteristics in ocean-atmosphere cycle,
        such as PWAT and RAIN.

        In addition, we also collect the historical Niño3, Niño4 and Niño3.4 indexes data from China Meteorological
Administration National Climate Centre (https://cmdp.ncc-cma.net/). We pick up the records from 2014.1 to 2020.8 for the
        result analysis of following experiments.

        The major active region of ENSO is concentrated in the tropical Pacific, so we crop the multivariate data with the
        region (N40°-S40°, E160°-W90°) as the geographic boundaries of ENSO-ASC, which covers Niño3 and Niño4 regions. The
        reanalysis data has the size (40, 55) for every single-month variable, and the remote sensing data has the size (320, 440). In
order to unify and improve dataset quality, we use Bicubic interpolation (Keys, 1981) to enlarge the reanalysis data by 8×
        magnification and soft-impute algorithm (Mazumder et al., 2010) to fill up missing values in both datasets. We train the
        model first on the whole reanalysis dataset and subsequently on the remote sensing dataset from 1997.12 to 2012.12 for fine-
        tuning. 2014.1 to 2020.8 in remote sensing dataset is considered as validation set. There is a one-year gap between fine-
        tuning set and validation set to reduce the possible influence of oceanic memory.

**4.2 Experiment Setting**

        We train and evaluate the ENSO-ASC on a high-performance server. Based on our proposed model, some hyper parameter
        settings are determined based on the existing calculation resources as following: $K = 4$, $d_1 = d_2 = 16$, which is the optimal
        parameter combination after extensive experiments (This process has been ignored because it is not the focus in this paper).
        Adjacency matrix $A$ and corresponding Laplacian matrix $\tilde{L}$ are designed as Section 3. All the following analyses are based
on the stable results through extensive experiments.

        We evaluate the ENSO-ASC from three aspects. Firstly, according to our proposed ENSO forecast formalization in Eq.
        (5) to Eq. (7), there are several factors that may influence the performance from the perspective of model structure: the input
        sequence length $M$, the multivariate coupler $coupler(\cdot)$, the attention weights $\alpha$ and $\beta$, and the benefits of transfer training.
        We design some comparison experiments to investigate the model performance and determine the optimal model structure.
A comparison with the other state-of-the-art models is also included. Secondly, we evaluate the ENSO forecast skill of the
        ENSO-ASC from the meteorological aspects in Eq. (5) to Eq. (7): the contributions of different input physical variables $V$ in





the pre-designed coupling graph $G$, the effectiveness of forecast lead time in IMS strategy, the forecast skill with different start calendar month $scm$, and the spatial uncertainties in a longer-term forecast. Finally, we forecast the real-world ENSO over the validation period and compare our results with the observations.

## 4.3 Evaluation of Model Performance

### 4.3.1 Influence of the Input Sequence Length

Sequence length $M$ is very important for forecasting model, due to the rich spatial-temporal contained in it. In general, the longer sequence length $M$, the better the ENSO forecast skill. However, it will increase the computational burden of the deep learning model and strengthen the quality requirements for the collected geoscience data, especially under such a high resolution of our model. Therefore, the balance between forecast performance and efficiency must be considered. Our proposed model uses IMS to forecast the ENSO evolution in the next future month, so we gradually increase the sequence length $M$ to detect the changes in forecast skill. Figure 10 displays the results. As the sequence length gradually increases, two metrics grow better. When the sequence length is greater than 3 months, the growth rate slows down, while the sequence length is less than 3 months, the forecast skill increase rapidly.

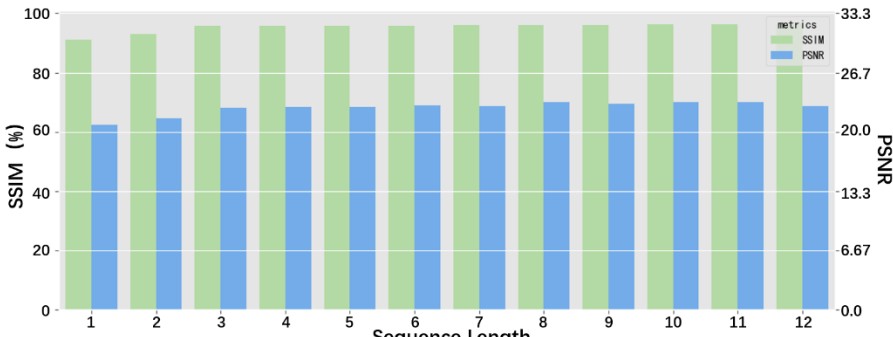

**Figure 10: The performance of the ENSO-ASC when the input sequence length increases under IMS forecast strategy.**

It is obvious that the increase of sequence length cannot lead to an unlimited improvement in forecast skill. In ENSO-ASC, making predictions with the previous 3 months' multivariate data is a more efficient choice. In fact, lots of successful works imply that climate deep learning model does not require a longer input sequence to make skilful predictions, such as using previous 2 continuous time-step data to estimate the intensity of tropical cyclone (Zhang et al., 2019a), using previous 3-month SST and wind data to predict ENSO evolution (Ham et al., 2019). A long-term temporal sequence must contain strong trends and periodicities, but the underlying chaos is more dominant, which seriously hinders the prediction. The subsequent experiments will all apply the historical 3-month multivariate sequence ($M = 3$) as model input data.



### 4.3.2 Benefits of the Transfer Learning

For the all model training, we use the transfer learning approach to overcome the insufficient sample challenge and obtain the optimal trained model. More specifically, we first train the model on reanalysis dataset with 90 epochs and subsequently on remote sensing dataset until total convergence (about 110 epochs). Here, 90 epochs are enough for training the ENSO-ASC on the reanalysis dataset until convergence, because the interpolated reanalysis data is smoother and lack of details, which leads to an easy-training. The original remote sensing dataset contains much more detailed high-level climate

information, so the model need more epochs (such as 200 epochs) to train.

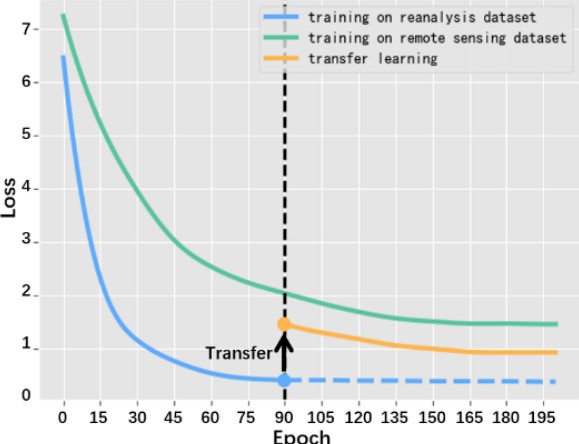

**Figure 11: The loss changes when training with only reanalysis dataset (blue line), with only remote sensing dataset (green line) and transfer learning on these two datasets in order (black arrow and yellow line after 90 epochs).**

We also make comparative experiments by only training our model on remote sensing dataset. We set the input

sequence length at 3 and train the model repeated times to obtain a stable result. The averaged loss is depicted in Fig. 11. We can see that when training with the reanalysis dataset, the loss drops quickly, while training with the remote sensing dataset, the model converges slowly and the loss is large. After using transfer learning approach, the loss on remote sensing dataset are improved at least 15%, which demonstrates that the systematic errors of ENSO-ASC are indeed corrected to some extent.

Comparing with remote sensing dataset, training with reanalysis dataset always yields a much smaller loss. It is due to

the smoothness and lack-of-details of the reanalysis dataset as above mentioned, which leads to the model can learn the characteristics more easily. However, the high-resolution remote sensing dataset reflects the real-world conditions more accurately, which contains more comprehensive and nonlinear details and amplitudes under a high resolution. If we have enough remote sensing data, the forecast skill will be further improved.

### 4.3.3 Effectiveness of the Multivariate Air-sea Coupler

We subjectively incorporate priori ENSO coupled interactions into the graph-based multivariate coupler and select 6 critical physical variables as the predictors of the ENSO-ASC. The formalization not only treats each physical variable as a separate





individual, but emphasizes the nonlinear connections between them. However, it is not clear whether such graph formalization is the reason for the improvement of ENSO forecast performance. In order to validate the effectiveness of our proposed formalization, we design two other deep learning couplers for ENSO forecast with the same datasets and transfer

learning, and then compare the performance with the original model.

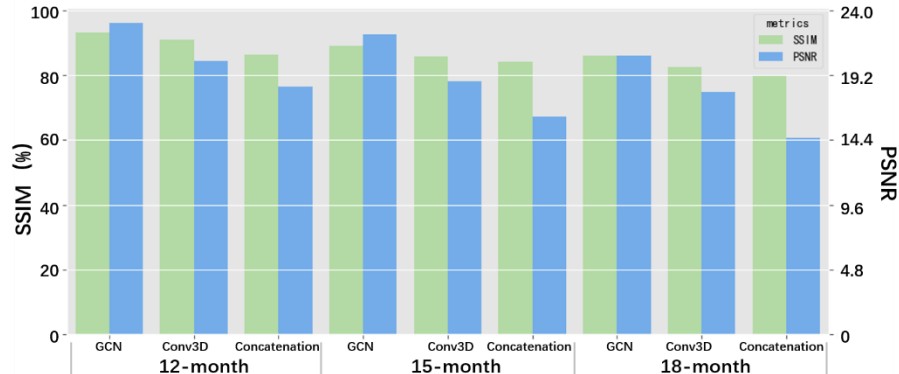

**Figure 12: The performance of the ENSO-ASC when replacing the multivariate air-sea coupler with other deep learning structures.**

The first coupler is to replace GCN with a dual-layer 3D-convolution block, which treats all variables as a whole system

and ignore the specific directions and neighbour-orders in coupled interactions between them. The second coupler just replaces the GCN with the concatenation of features from multivariate encoders, which treats the multiple variables as a channel stacking (or data overlay) and simply extracts global features of them together (the concatenated multivariate features are passed into multivariate decoders directly). The results are illustrated in Fig. 12. Obviously, our graph formalization achieves the best performance with measurements of *SSIM* and *PSNR*, the Conv3D coupler is slightly worse.

The results indicate that using a graph to simulate multivariate interactions is a more reasonable approach, which can learn more ENSO-related dynamical interactions and underlying physical processes than other formalizations. Besides this, the comparative experiments also exhibit some inspiration: When building a climate forecast deep learning network that incorporates physical mechanisms, it is necessary to select a suitable structure to support and express specific mechanisms mathematically.

We design a full-connected adjacency matrix $A$ in GCN coupler, which means we consider that all physical variables have interactions with each other. Conv3D also entirely extracts the features of all variables. Under this circumstance, why GCN coupler has better performance than Conv3D coupler? From the perspective of mathematics, GCN will consider the pair-wise coupling between variables and learn the features of every coupled interaction according to the hidden nonlinearity in samples individually as Eq. (12). But the Conv3D coupler inherits the characteristics of the global-share and local-connect

parametric designs in classic convolution, which rather treats different variables as different channels in images, lacking more detailed coupled information.





### 4.3.4 Effects of Attention Weights

We customize two attention components in the model to overcome the problem of negative effects of temporal memories and multiple variables. Here, we analyze the influences of two proposed self-supervised attention weights, the results are
illustrated in Fig. 13. The results suggest the prediction skill will decline when one of the attention techniques removed. More specifically, the reduction of performance is larger when the multivariate attention is removed for shorter forecasts (less than about 9 months), or when the temporal attention is removed for longer forecasts (more than about 15 months). It is because of higher multivariate correlations and lower temporal non-stationarity in the short term. But the temporal memory effects dominant in the long term.

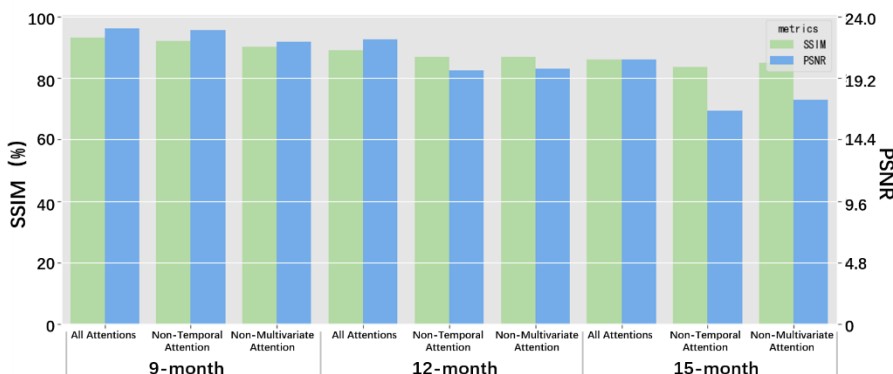


**Figure 13: The performance of the ENSO-ASC when removing one of the attention components.**

In fact, due to the self-supervised attentive weights $\alpha$ and $\beta$, even when the multivariate graph and model structure are fixed, the prediction skills will not change too much with the difference start calendar months and variable combination (See Section 5.2.3). Because of the IMS forecast strategy, the prediction with the different start calendar months will share one
common model. If these two components are unset, the model will not be able to distinguish the contributions of multivariate oceanic memories in different situations adaptively, seriously misleading the forecast.

Indeed, it may be a better choice for ENSO forecast to establish and filter the optimal model individually for different start calendar months with various variables, but it also means higher resource and time consuming. These two attentive weights $\alpha$ and $\beta$ can reasonably prune the model within the acceptable range of prediction errors, which is more like the
traditional dynamic forecasting manner. In operational forecasts, separate modelling methods can be used to pursue higher accuracy and skills.

### 4.3.5 Comparison with Other State-of-the-art ENSO Deep Learning Models

We compare the ENSO-ASC with other state-of-the-art data-driven ENSO forecast models, including: (1) Convolutional Neural Network (Encoder-decoder structure with 12 layers, which has the same trainable layer number with our model); (2)
Long Short-Term Memory Network (6 LSTM layers and final fully connect layer); (3) ConvLSTM Network (CL-6 means a





6-layer structure, CL-12 means a 12-layer structure); (4) 3D-Convolution coupler to simulate multivariate interactions as mentioned above (Conv3D). In order to ensure the fairness of the comparison, we utilize the same input physical variables, training/validation datasets and training criteria for the above models, then train them via transfer learning with plenty of epochs to achieve their optimal performances. Table 2 displays the comparative results with 12-/15-/18-month forecast. In

general, the forecast models considering ENSO spatial-temporal correlations (e.g. ConvLSTM, Conv3D) outperform the basic deep learning models (e.g. CNN, LSTM), which implies that the complicated network structures can mine the sophisticated rules deeply hidden in long-term ENSO forecast more effectively. As the lead time increases, the performance of models gradually decreases. However, the ENSO-ASC still maintains high accuracy and is always better than other models with an increment of about 5%, which indicates the superiority of our model.

**Table 2: Performance comparisons with other state-of-the-art deep learning models**

| Model | 12-month SSIM / PSNR | 15-month SSIM / PSNR | 18-month SSIM / PSNR |
|---|---|---|---|
| *CNN* | 86.32 / 17.47 | 83.97 / 14.40 | 79.85 / 12.20 |
| *LSTM* | 88.57 / 18.08 | 84.19 / 15.41 | 81.59 / 13.58 |
| *CL-6* | 88.70 / 18.37 | 84.36 / 16.25 | 81.73 / 14.04 |
| *CL-12* | 89.78 / 19.45 | 84.74 / 17.34 | 82.03 / 15.37 |
| *Conv3D* | 90.93 / 20.16 | 85.59 / 18.01 | 82.50 / 15.98 |
| *ENSO-ASC* | 92.65 / 22.05 | 90.31 / 20.97 | 87.53 / 17.17 |

Considering the calculation ingredients of *PSNR* and *SSIM* according to Eq. (11): *PSNR* only contains *MSE*, which is the metric for individual grids, while *SSIM* measures the spatial characteristics and distributions of two patterns from many correlation coefficients, which can represent a measurement for evolution tendency and physical consistency to some extent. Based on the above analysis, Table 2 also indicates that the forecast skill of the ENSO-ASC exhibits a big improvement over

other models, the *SSIM* of which is much better, especially in the longer lead time. In addition, the ENSO-ASC pay more attention on the detailed spatial distributions, which is beneficial for the further analysis of ENSO dynamical mechanisms.

On the other hand, the ENSO-ASC is the first attempt to forecast ENSO at such a high resolution (0.25 °). Despite the difficulty of training increases, our model still achieves good results. Interestingly, though the ENSO-ASC involves the most trainable parameters, its convergence epoch is one-fifth on average of other forecast models.

**4.4  Analysis of ENSO forecast skill**

**4.4.1 Contributions of Different Predictors to the Forecast Skill**

The superiority of our proposed model derives from the graph formalization, and the special multivariate coupler can effectively express the processes of synergies between multi-physical variables. From another perspective, the improvement of the forecast skill is not only benefited from graph formalization, but also due to the utilization of multiple variables highly

related to ENSO compared to using limited variable to predict ENSO as previous works. For ENSO forecast, SST is definitely the most critical predictor. Besides SST, other variables have different contributions to the forecast results.





Therefore, we design an ablation experiment by removing one predictor from our proposed model and detect the reduction of forecast skill.

**Table 3: Model performance when one existing variable removed or one extra variable added**

| Removed variable | 12-month SSIM / PSNR | 15-month SSIM / PSNR | 18-month SSIM / PSNR |
|---|---|---|---|
| - | 92.65 / 22.05 | 90.31 / 20.97 | 87.53 / 18.17 |
| *RAIN* | 91.46 / 21.34 | 88.74 / 18.32 | 85.86 / 17.35 |
| *CLOUD* | 91.53 / 21.65 | 88.81 / 18.54 | 85.93 / 16.16 |
| *VAPOR* | 91.52 / 21.65 | 88.82 / 18.53 | 85.92 / 16.16 |
| *UWIND* | 90.08 / 20.93 | 87.03 / 17.81 | 83.72 / 13.58 |
| *VWIND* | 91.47 / 21.62 | 88.65 / 18.42 | 85.31 / 15.07 |
| Added variable | 12-month SSIM / PSNR | 15-month SSIM / PSNR | 18-month SSIM / PSNR |
| *Surface Pressure* | 92.74 / 22.13 | 90.33 / 20.99 | 87.64 / 17.26 |
| *Surface Air Temperature* | 92.75 / 22.15 | 90.40 / 21.07 | 87.71 / 17.25 |
| *upper ocean heat content* | 92.98 / 22.14 | 90.45 / 21.10 | 87.79 / 17.34 |

Table 3 (above) shows that when a variable is removed from the input of the deep learning model, the ENSO forecast skill will be reduced. More specifically, when the zonal weed speed (UWIND) is removed, the reduction is the largest. From the perspective of ENSO physical mechanism, zonal wind anomalies (ZWA) always play a necessary role and are even considered as the co-trigger or driver of ENSO events. As an atmospheric variable, ZWA often gives a direct feedback on oceanic varieties with a shorter response time than oceanic memory. ENSO-ASC uses historical 3-month multivariate data to

predict ENSO evolution, which is a quite short sequence length. Under such sequence length, wind speed (including u-wind and v-wind) has a relatively high correlation with SST. In addition, RAIN is another variable that slightly affects the forecast. This is because the precipitation process has a straightforward contact with the sea surface, and the energy transfer is easier.

In addition, we also add other variables into the multivariate coupler (surface air temperature/surface pressure/ocean heat content), then analyze the forecast skill change of the ENSO-ASC (the description of them are in the Table 3), the

results exhibit little improvements on the performance, shown as Table 3 (below). There are two possible reasons for this result: Firstly, the contributions of the extra added variables to the evolution of ENSO is not as obvious as the existing chosen variables (such as pressure or air temperature), while the multivariate graph with existing variables can almost describe a relatively complete energy loop in Walker circulation. So the effect of extra variables for model is small. Secondly, the concept of extra variables may be overlapped with the existing variables for the ENSO process. For example,

the upper ocean heat content (vertically averaged oceanic temperature in the upper 300m) has more capabilities to describe the changes in the dynamic process of ENSO (Ham et al., 2019; McPhaden, 2003), which has been formalized in the



recharge oscillator theory of ENSO (Jin, 1997) and has been verified against observations (Meinen and McPhaden, 2000). The calculation of this variable contains the region of SST, so the effect of the extra introduction of upper ocean heat content will be weakened. From the perspective of dataset cost, the 6 chosen variables in the model can be obtained more easily with

a quite high quality, and the model performance has reached an acceptable high level. While it can be reasonably speculated that the forecast skill of the coupler will be improved if considering more meteorological mechanisms with more crucial variables.

### 4.4.2 Analysis of Effective Forecast Lead Month

The accuracy of long-term prediction is the most crucial issue for meteorological research. In ENSO events, though the

periodic dominants the amplitude, the intrinsic intensity and duration often contribute very large uncertainties. Therefore, over the validation period, we make predictions with multiple lead times and calculate the corresponding Niño indexes from the forecasted SST patterns. The correlations between forecasted Niño indexes and the official records are depicted in Fig. 14. As the lead times increase in 24 months, the correlation skill slowly decreases. It is worth noting that when the lead time is from 10 to 13 months, the decline of these three indexes forecast skills slow down a little. This is because that the periodic in

ENSO events become stronger after a 1-year iteration of IMS strategy. These results demonstrate that the ENSO-ASC can provide reliable predictions up to at least 18 to 20 months on average (with correlations over 0.5). Within 6-month lead time, the correlation skill is over about 0.78, and from 6- to 12-month lead time, correlation skill is over about 0.65.

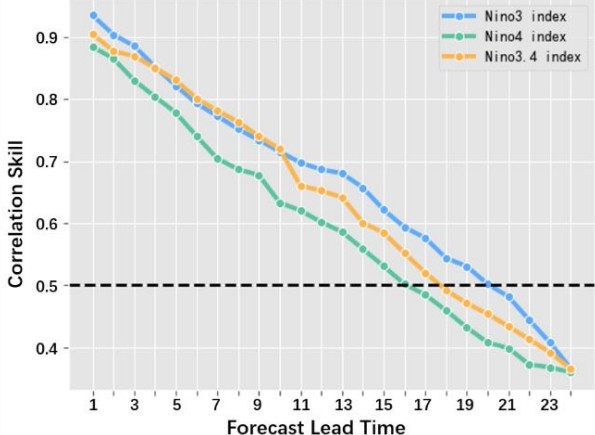

**Figure 14: The correlation skills between the forecast results of the ENSO-ASC and real-world observations on three Niño indexes**
**with the forecast lead time increasing over the validation period.**

In addition, the forecast skill for Niño3 index and Niño3.4 index is a little higher than that of Niño4 index. It indicates that our model has higher forecast skill for the EP-El Niño events (the active area of SST anomalies is majorly over the eastern tropical Pacific Ocean) than the CP-El Niño events (the active area of SST anomalies is majorly over the western and central tropical Pacific Ocean). This is because that the input area of the model majorly covers the entire tropical Pacific,

which can be considered as the sensitive area for EP-El Niño events. As for lower forecast skills of CP-El Niño events,





extratropical Pacific or other oceanic regions may have stronger impacts on the western-central equatorial Pacific (Park et al., 2018).

### 4.4.3 Temporal Persistence Barrier with Different Start Calendar Months

Deep learning model can extend the effective lead time of ENSO forecasts, which means it can raise the upper limitation of
ENSO prediction to some extent. From the perspective of IMS strategy, if a well-trained model can predict next-month SST perfectly (in other words, with a very low predict error), the model can forecast long enough iterations theoretically. However, our proposed model is affected by a variety of factors, which leads to performance degradation.

One of the disadvantages in IMS strategy is that: Once a relatively large forecast error shows up in a certain iteration, such forecast error will be continuously amplified in subsequent iterations. In ENSO forecast, such forecast skill decline is
called persistence barrier and usually happens in spring (i.e. spring predictability barrier, SPB) (Webster, 1995; Zheng and Zhu, 2010). SPB limits the long-term forecast skill in not only numerical models but some other statistical models (Kirtman et al., 2001). For further investigating the reason of performance degradation, we first make continuous predictions over the validation period from different start calendar months with different lead time, and then calculate the correlations between the calculated Niño3.4 index with the official records. Figure 15 shows the results.

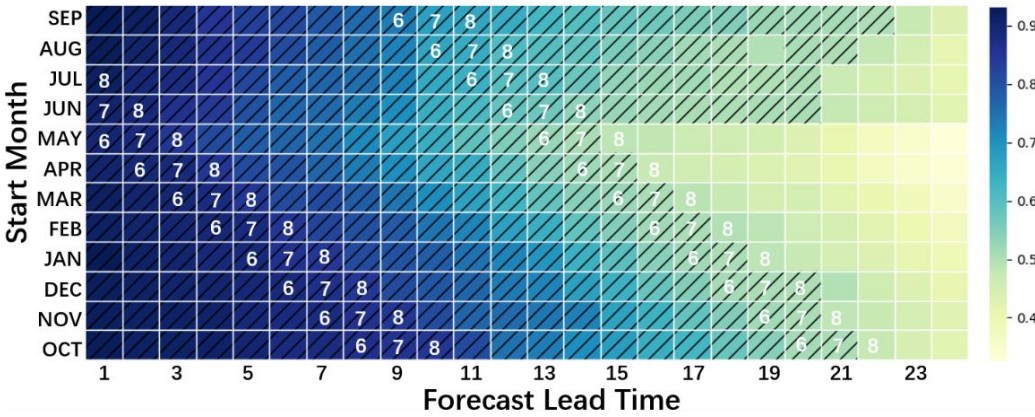


**Figure 15: The correlation skill heat map between the forecast results of the ENSO-ASC and real-world observations on Niño3.4 index with different start month over the validation period. The hatching cells represent the correlation over 0.5, and the white numbers on the cells mean the forecast months.**

In Fig. 15, the darker the cells' color, the higher the correlations between the forecast and observation, the higher the
forecast skill. The hatching cells represent the correlations exceeding 0.5. Overall, making ESNO forecast with start calendar months MAM (March, April and May) is not very reliable, while the long-term forecast of ENSO is easier with start months JAS (July, August, and September). In addition, there exist two obvious color change zones among the whole cells, which means the correlations dropped significantly in such zones (cell color becomes lighter), and both of them occurs in the months JJA (June, July and August) depicted as the white numbers on the cells. The first zone reduces the correlations by



about 0.03, and the second zone make the reduction by about 0.06. It demonstrates that when making forecasts through the months JJA, the ENSO predictions tend to be much less successful. This is why the model exhibits more skilful forecast with the start months JAS, which avoids forecasting through the months JJA as much as possible and preserves more accurate features during iterations, resulting in a relatively long efficient lead time. Analogy to SPB in traditional ENSO forecast, our proposed deep learning model exists forecast persistence barrier in boreal summer (JJA). This may be because that our real-

world dataset contains more frequency CP-El Niño events after 1990s (Kao and Yu, 2009; Kug et al., 2009), which is significantly impacted by summer predictability barrier (Ren et al., 2019; Ren et al., 2016). At the same time, it also implies that there are still forecast obstacles we need to circumvent in the deep learning ENSO forecast model, more unknown key factors need to be considered and explored, such as more variables, larger input regions, more complex mechanisms, etc. It will make great progress from building deep learning model based on prior meteorological knowledge.

**4.4.4 Spatial Uncertainties in Longer Lead Time**

In ENSO forecast, the areas where the forecast uncertainties occur are usually not randomly distributed, and such areas should be focused more in observation. Over the validation period, we make 12-month and 18-month forecasts, and then compare the forecast results with observation. More specifically, we calculate covariance between forecast sequence $\hat{s}$ and observation $s$ for every grid point and combine them as a spatial heat map.

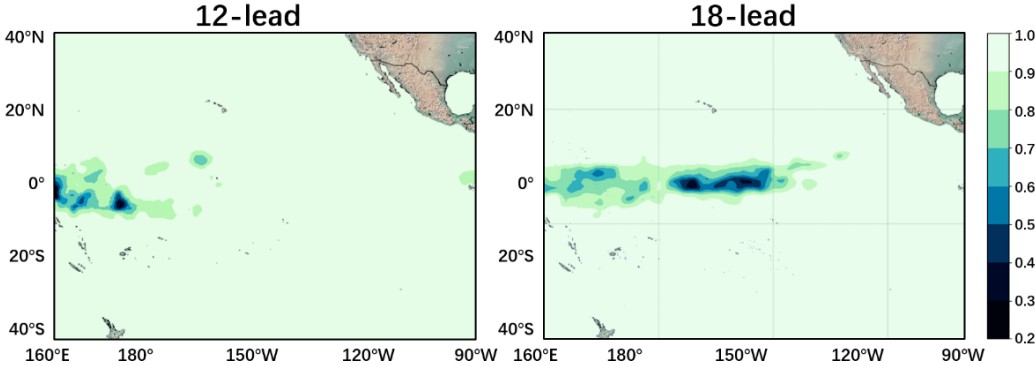


**Figure 16: The spatial covariance heat map between the forecast results of the ENSO-ASC and real-world observations with 12 and 18 lead month over the validation period.**

The results are shown in Fig. 16. The spatial uncertainties first show up over the western equatorial Pacific, then as the lead time increases, the uncertainty area gradually expands eastward to the central equatorial Pacific. It indicates that the

Niño3 and Niño3.4 indexes both have very high forecasting skills in a short forecast lead time (See Fig. 14 before 12-month forecast), while the forecast for the central equatorial Pacific is gradually disturbed in a longer lead time, which leads to a rapid reduction of forecast skill for Niño3.4 and Niño4 indexes shown as Fig. 14 after 15-month forecast. This illustrates that the area with the largest forecast uncertainty is generally the area with the most active SST evolution in ENSO. Besides this, another possible reason is that the input multivariate region is confined in the Pacific, but the ocean-atmosphere coupled





interactions in the western tropical Pacific can be profoundly influenced by extratropical Pacific areas or other ocean basins
as mentioned above. Therefore, our proposed model has relatively weak ability to capture the development of SST over the
western-central equatorial Pacific.

### 4.5 Simulation of the Real-world ENSO Events

Since the 21st century, the occurrences of ENSO are more and more frequent, especially the duration and intensity of ENSO
have largely changed. For example, many numerical climate models failed to forecast the 2015/2016 super El Niño. We
simulate several ENSO events during validation period and compare the forecast results with real-world observations. As
mentioned above, wind (u-wind and v-wind) is also a relatively important and sensitive predictor in the ENSO-ASC for
ENSO forecast. Therefore, we make an 18-month forecasts and majorly trace the evolutions of SST and wind (u-wind and v-
wind). Note that all of the following patterns describe the evolutions of SST and wind anomalies (remove the climatology
from the forecasted SST and wind patterns) for more convenience.

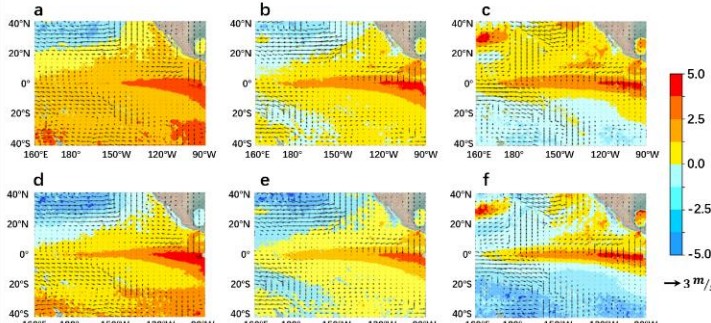

**Figure 17: The growth phase of SST anomalies in 2015/2016 super Niño event from 2015.4 to 2015.6. a-c are the forecast results of the ENSO-ASC and d-f are real-world observations.**

Figure 17 display the evolutions of SST and wind anomalies in growth phase of 2015/2016 super El Niño event from
2015.4 to 2015.6, where a-c sub-figures are forecasts and d-f sub-figures are observations. Fig. 18 display the peak phase
from 2015.9 to 2016.2, where a-f sub-figures are forecasts and g-l sub-figures are observations. These two results are both
with the prediction start time of 2015.1. During the growth phase, the warming SST anomalies first show up over the eastern
tropical Pacific Ocean, which reduces the east-west gradient of SST. At the meanwhile, the westerly wind anomalies over
the western-central equatorial Pacific further enhances the SST anomalies over central-eastern equatorial Pacific and
weakens the Walker circulation (Fig. 17 a-c). The SST and wind anomalies trigger the Bjerknes positive feedback together,
which cause SST anomalies to be continuously positively amplified. During the peak phase, in addition to the local
evolutions of the equatorial Pacific SST anomalies, there are obvious warm SST anomalies over the northeast subtropical
Pacific near Baja California induced by the extratropical atmospheric varieties (Yu et al., 2010; Yu and Kim, 2011), which
gradually propagates south-westward and merge with the warm SST anomalies over the central equatorial Pacific (Fig. 18 a-
d). In conclusion, the ENSO-ASC can track the large-scale oceanic/atmospheric varieties steadily and successfully predicts



the very-high-intensity and very-long-duration, while many dynamic or statistical models fail. At the same time, out proposed model makes the prediction at the beginning of the calendar year and produce a quite low prediction error, which demonstrates that the model can overcome or eliminate the negative impacts of SPB to some extent.

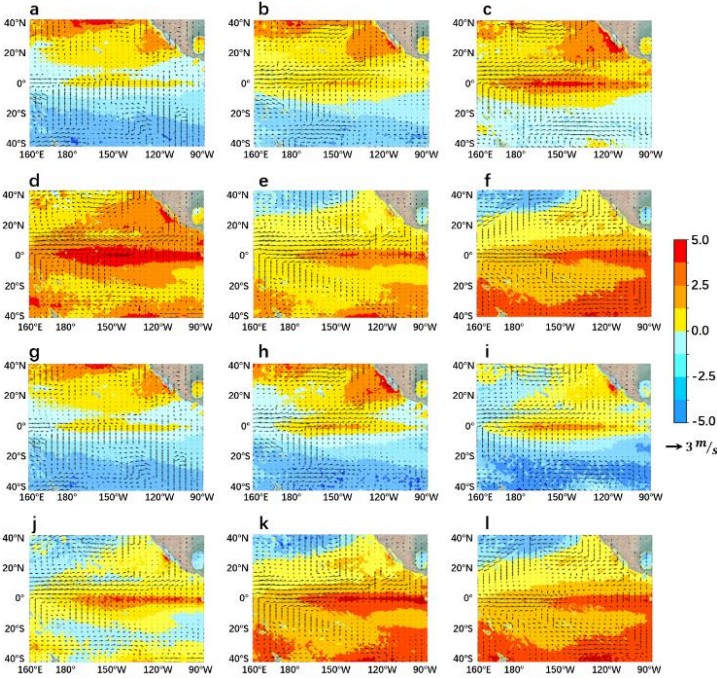

**Figure 18: The peak phase of SST anomalies in 2015/2016 super Niño event from 2015.9 to 2016.2. a-f are the forecast results of the ENSO-ASC and g-l are real-world observations.**

From this real-world simulation, our model takes full advantage of multivariate coupled interactions and extract more nonlinear characteristics in the ENSO evolution, which has been tracking the developments steadily.

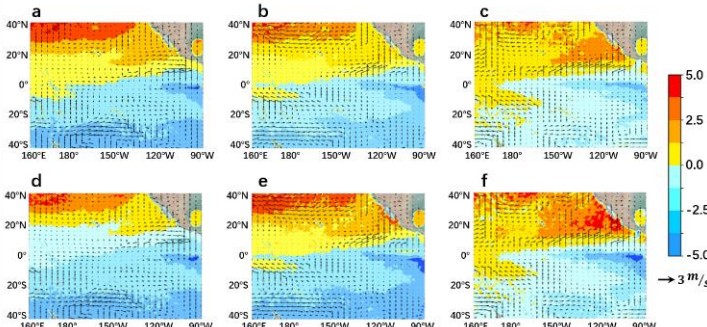

**Figure 19: The same with Fig. 17 but for the growth phase of SST anomalies of 2017 weak La Niña event from 2017.9 to 2017.11.**

Besides the super El Niño event, the ENSO-ASC also has higher simulation capabilities for weaker nonlinear unstable evolutions. In reality, neutral or weaker events actually account for most of the time. Judging from the saliency of the extracted features, weaker events may contain "more mediocre and fuzzy" characteristics, which leads to some difficulties





for such predictions to some extent. For example, it is much easier to overestimate or underestimate their intensities.

Therefore, we choose a hindcast over the validation period. Fig. 19 shows the peak phase of a weak La Niña event from 2017.9 to 2017.11 with the prediction start time of 2016.9, where a-c sub-figures are forecasts and d-f sub-figures are observations. From its evolution, there are negative SST anomalies over the eastern equatorial Pacific and easterly wind anomalies in the western tropical Pacific Ocean, which will enhance the Walker circulation. In addition, Bjerknes positive feedback is the dominant factor favouring the rapid anomaly growth in this simulation.

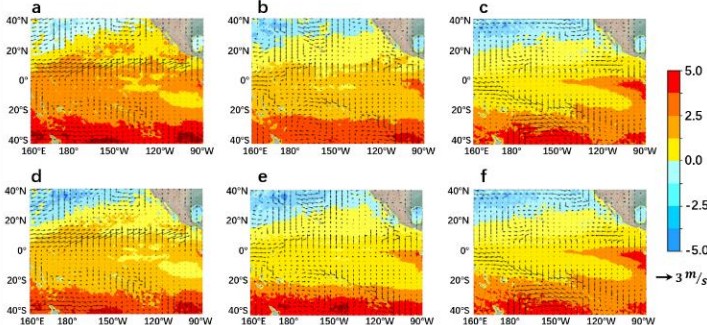


**Figure 20: The same with Fig. 17 but for the neural SST evolution in 2020.1 to 2020.3.**

    Another ENSO forecast limitation is to predict the neural year as the event of El Niño (or La Niña), which is also known as false alarm rate. Figure 20 displays the neutral event from 2020.1 to 2020.3 with the prediction start time of 2019.1 (Its ONI has not yet reached the intensity of El Niño). After calculating the corresponding Niño index, we can determine that

the ENSO-ASC can accurately avoid this situation and credibly reflect the real magnitude of the development of important variables such as SST. We have also verified the case in 2014 and the result is consistent with the facts. Many operational centres erroneously predicted that an El Niño would develop in 2014, but it did not.

    The forecasted SST and wind anomaly patterns have a consistent intensity and tendency with the observations. Why our model can achieve better forecast skills in a variety of situations? Because our proposed deep learning coupler absorbs the

sophisticated oceanic and atmospheric varieties, whose deep and intricate structure can almost simulate the air-sea energy exchange simultaneously, while traditional geoscience fluid programming in numerical climate model usually applies interval flux exchange, blocking the continuous interactions.

## 5 Discussions and Conclusions

ENSO is a very complicated air-sea coupled phenomenon, the life-cycle of which is closely related to the large-scale

nonlinear interactions between various oceanic and atmospheric processes. ENSO is one of the most critical factors that cause extreme climatic and socioeconomic effects. Therefore, meteorological researchers are starting up to find more accurate and less consuming data-driven models to forecast ENSO, especially deep learning methods. There are already many successful attempts that extend the effective forecast lead time of ENSO up to one and a half year. They all extract the



rich spatial-temporal information deeply hidden in the historical geoscience data.

However, most of the models use limited variables even single variable to predict ENSO, ignoring the coupled multivariate interactions in ENSO events. At the same time, the generic ENSO deep learning forecast models seem to have reached the performance bottleneck, which means deeper or more complex model structures can neither extend the effective forecast lead time nor provide a more detailed description for dynamical evolutions. In order to overcome these two barriers, we subjectively incorporate priori ENSO knowledge into the deep learning formalization and derive hand-crafted features

into models to make predictions. More specifically, considering the multivariate coupling in the Walker circulation related to ENSO amplitude, we select 6 indispensable physical variables and focus on the synergies between them in ENSO events. Instead of simple variable stacking, we treat them as separate individuals and ingeniously formulate the nonlinear interactions between them on a graph. Based on such formalization, we design a multivariate air-sea coupler (ASC) by graph convolution mathematically, which can mine the coupled interactions between every physical variable in pairs and perform

the multivariate synergies simultaneously.

   We then implement an ENSO deep learning forecast model (ENSO-ASC) with the encoder-coupler-decoder structure, and two self-supervised attention components are also integrated. The multivariate time-series data is passed to the encoder to extract spatial-temporal features respectively; Then the multivariate features are aggregated together for interactions in the multivariate air-sea coupler; Finally, the coupled features are divided separately and the corresponding feature of a certain

variable is restored to forecast patterns in the decoder respectively. IMS strategy is applied to make prediction, which is a more stable forecast way. We use transfer learning to provide a better model initialization and overcome the problem of observation sample-lacking. The model is first trained on the reanalysis dataset and subsequently on remote sensing dataset. After constructing the model structure, we design extensive experiments to investigate the model performance and ENSO forecast skill. Some successful simulations in the validation period are also provided. Some conclusions can be summarized

as follows:

   1) According to the forecast model described in Eq. (5) to Eq. (7), we adjust the model settings of the input sequence length $M$, the multivariate coupler $coupler(\cdot)$, the attention weights $\alpha$ and $\beta$, the transfer training, and then investigate the performance change. The optimal input sequence length of the model is 3 according to the trade-off between forecast skill and speed. It implies that the ENSO deep learning forecast model does not need a relatively long-

sequence input. Although the long sequence contains rich tendency and periodic of ENSO events, the meteorological chaos is more dominant, which seriously hinders the prediction. Transfer learning is a practical way. Training the model on the reanalysis dataset and subsequently on the remote sensing dataset can effectively reduce the system forecast errors by at least 15%. When replacing the graph-based multivariate air-sea coupler in ENSO forecast model with other deep learning structures, the forecast skill drops obviously. This demonstrates that the graph formalization is

a powerful expression for simulating underlying air-sea interactions and corresponding ENSO forecast model with novel multivariate air-sea coupler can forecast more realistic meteorological details. This also demonstrates that it is critical to choose suitable deep learning structures to incorporate prior climate mechanisms for improving forecast skills.





The self-supervised attention components are also promising tools to eliminate the performance reductions caused by different physical variables and start calendar month. In addition by comparing with other state-of-the-art ENSO forecast models, the ENSO-ASC achieves at least 5% improvement in *SSIM* and *PSNR* of long-term forecasted SST patterns.

2) By performing the ablation experiment, the forecast skill drops significantly when removing the zonal wind speed from the model input, which is because it is a co-trigger of Bjerknes positive feedback in ENSO events and gives a direct feedback on oceanic varieties with a shorter lag time. Adding extra variables can slightly improve the performance, especially the ocean temperature heat content, which is the concept and calculation extension of the original SST. By tracing the upper limitation of forecast lead time, the ENSO-ASC can provide a reliable high-resolution ENSO forecast up to at least 18 to 20 months on average judging from the correlation skill of Niño indexes greater than 0.5. Within 6-month lead time, the correlation skill is over about 0.78, and from 6- to 12-month lead time, correlation skill is over about 0.65. The corresponding correlation skills decline slowly from 10- to 13-month lead time, and then declined rapidly. This is because of the stronger periodic in ENSO events after a 1-year iteration of IMS strategy. At the same time, the different forecast start calendar months is influential for the performance. The temporal heat map analysis shows that an obvious skill reduction usually shows up in JJA and produces a boreal summer persistence barrier in our model. In addition, from the spatial uncertainty heat map, our model exhibits larger forecast uncertainties over the SST active region, especially western-central equatorial Pacific. Such spatial-temporal predictability barriers are widely present in dynamic or statistical models, but the ENSO-ASC effectively prolongs the forecast lead time and reduces corresponding uncertainties to a large extent.

3) Some successful simulations exhibit the effectiveness and superiority of the ENSO-ASC. We make real-world ENSO simulations during the validation period by tracing the evolutions of SST and wind anomalies (u-wind and v-wind). In the forecasted El Niño (La Niña) events, the sea-air patterns clearly display that the positive (negative) SST anomalies first show up over the eastern equatorial Pacific with westerly (easterly) wind anomalies in the western-central tropical Pacific Ocean, which induces the Bjerknes positive feedback mechanism. As for 2015/2016 super El Niño, the ENSO-ASC captures the strong evolutions of SST anomalies over the northeast subtropical Pacific in the peak phase and successfully predicts its very-high-intensity and very-long-duration, while many dynamic or statistical models fail. ENSO-ASC can also credibly reflect the real situation and reduce the false alarm rate of ENSO such as in 2014. In conclusion, our model can track the large-scale oceanic and atmospheric varieties and simulate the air-sea energy exchange simultaneously. It demonstrates that the multivariate air-sea coupler effectively simulates the oscillations of Walker circulation and reveals more complex dynamic mechanisms such as Bjerknes positive feedback.

Meteorological research does not only pursue skilful models and accurate forecasts, but requires a comprehensive understanding of the potential dynamical mechanisms. The extensive experiments demonstrate that the ENSO forecast model with a multivariate air-sea coupler (ENSO-ASC) is a powerful tool for analysis of ENSO-related complex mechanisms. Thus we will use it in the ENSO predictability research in depth. We will identify the optimal precursor by

involving more physical variables with informative vertical layers (such as the thermocline depth, and the ocean temperature heat content) considering more sophisticated dynamic mechanisms. Furthermore, we will carry out sensitivity investigation via our model, revealing the most significant temporal correlations and spatial uncertainties within multiple ENSO-related

variables, which can promote interdisciplinary research between computer science and meteorological geoscience.

**Code availability**

The source code of the ENSO-ASC 1.0.0 is available at https://doi.org/10.5281/zenodo.5081794.

**Data availability**

Thanks to NOAA/CIRES, Remote Sensing System, and China Meteorological Administration for providing the historical

geoscience data and analysis tools. (https://rda.ucar.edu/, http://www.remss.com/, https://cmdp.ncc-cma.net, last access: 8 Jul 2021).

**Author contribution**

All authors design the experiments carry them out. Bo Qin develop the model code and perform the simulations. Bo Qin and Shijin Yuan prepare the manuscript with contributions from all co-authors.

**Competing interests**

The authors declare that they have no conflict of interest.

**Acknowledgment**

This study is supported in part by the National Key Research and Development Program of China under Grant 2020YFA0608002, in part by the National Natural Science Foundation of China under Grant 42075141, in part by the

Fundamental Research Funds for the Central Universities under Grant 22120190207, and in part by the key project fund of Shanghai 2020 "science and technology innovation action plan" for social development under Grant 20dz1200702.

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
