# Peer review of "ENSO-ASC 1.0.0: ENSO Deep Learning Forecast Model with a Multivariate Air-Sea Coupler"

_Geoscientific Model Development, 2021_

## Author Comment (AC1)

Dear Editor and Reviewer:

Thank you very much for your insightful comments concerning our manuscript "ENSO-ASC 1.0.0: ENSO Deep Learning Forecast Model with a Multivariate Air–Sea Coupler" (ID: gmd-2021-213). Those comments are all very valuable and helpful for revising and improving our manuscript, we have studied comments carefully and have made revision. The point-by-point responses are as following:

**Comment 1:** 260: There's an $I_n$ in this formula but was not explained ahead. This is supposed to be an identity matrix, right? Please introduce it clearly.

**Response:** Thank you for spotting our crucial neglects in calculation description. We have updated the corresponding statements at **line 261** as the **blue** text below:

"*In practical implementation mathematically, we use graph Laplacian matrix L to normalize the energy flow of original adjacency matrix A as Eq. (11),* **where $I_n$ is an identity matrix with the order $n \times n$**. *L can be considered as the directions in which the excess unstable energy will propagate to other variables when the entire system is perturbed (such as external wind forcing).*"

**Comment 2:** Figure17-Figure20: The data seems to be standardized and should be mentioned on the figure.

**Response:** Thank you so much for your professional attitude and helping us find a mistake. Figure17-Figure20 are used to display the SST and wind anomalies patterns of the model results and observations, which are obtained by subtracting the corresponding climatology (climate mean state). However, we made a mistake. For example, for the SST anomalies patterns in July, we should subtract the climatology of SST in July, but we subtract the average of the climatology of SST in the whole year. By the way, the climatology in July is the average of the data in July of the recent 30 years. So, Figure17-Figure20 in the previous manuscript is wrong shown as Figure1-Figure4.

The corrected Figure17-Figure20 in manuscript are shown as Figure5-Figure 8, which have been updated at **line 665, 690, 695, and 710** in the latest manuscript. We also supplement the related statements at the line 635 as the **blue** text below:

"*Therefore, we make long-term forecasts and majorly trace the evolutions of SST and wind (u-wind and v-wind).* **Note that all of the following patterns describe the evolutions of SST and wind anomalies by subtracting the climatology (climate mean state) of that month (the recent 30-year monthly averaged SST and wind) from the forecasted SST and wind patterns.**"

[Figure]

**Figure 1: The growth phase of SST anomalies in 2015/2016 super Niño event from 2015.4 to 2015.6. a-c are the forecast results of the ENSO-ASC and d-f are real-world observations.**

[Figure]

**Figure 2: The peak phase of SST anomalies in 2015/2016 super Niño event from 2015.9 to 2016.2. a-f are the forecast results of the ENSO-ASC and g-l are real-world observations.**

[Figure]

**Figure 3: The same with Fig. 17 but for the growth phase of SST anomalies of 2017 weak La Niña event from 2017.9 to 2017.11.**

[Figure]

**Figure 4: The same with Fig. 17 but for the neural SST evolution in 2020.1 to 2020.3.**

[Figure]

**Figure 5: The growth phase of SST anomalies in 2015/2016 super Niño event from 2015.4 to 2015.6. a-c are the forecast results of the ENSO-ASC and d-f are real-world observations.**

[Figure]

**Figure 6: The peak phase of SST anomalies in 2015/2016 super Niño event from 2015.9 to 2016.2. a-f are the forecast results of the ENSO-ASC and g-l are real-world observations.**

[Figure]

**Figure 7: The same with Fig. 1 but for the growth phase of SST anomalies of 2017 weak La Niña event from 2017.9 to 2017.11.**

[Figure]

**Figure 8: The same with Fig. 1 but for the neural SST evolution in 2020.1 to 2020.3.**

**Comment 3:** A suggestion: In this paper, it is found that the best effect is to set the input sequence length as 3. This may be due to selecting the predictors with short memory (vapor, cloud). If predictors with long memory (such as heat content) are added, it may be more effective to set the length longer. Although Table 3 shows the prediction effect of the model with increased heat content data, the input sequence length is the same. This may be taken into consideration in a future study using global data.

**Response:** We thank the reviewer for this valuable insight very much. As the reviewer said, from our experiments in Section 4.3.1 *(Influence of the input sequence length)*, we found that the suitable input sequence length for the ENSO-ASC is 3 months according to the trade-off between the time-/resource-consuming and forecast skill when using 6 predictors (SST, u-wind, v-wind, rain, cloud, and vapor), in which 5 variables are related to the atmospheric processes with short memories. In the Section 4.4.1 *(Contributions of different predictors to the forecast skill)*, when we add heat content with long memory into the model, it is indeed necessary to re-investigate the optimal input sequence length by experiments in this manuscript. In fact, in continuous studies following this manuscript, the input length should be at least 6 months with 7 input variables (SST, u-wind, v-wind, rain, cloud, vapor and heat content) using globe data. While with the equatorial Pacific data and the input sequence length varying from 3 to 9 months, the change of forecast skill of ENSO-ASC is not much significant. Because the input region mainly covering the equatorial Pacific and most of the variables are with short memories in this manuscript, the input sequence length is still set as 3 despite adding heat content data into the model shown in Table 3. Let's look forward to our next manuscript following this manuscript.

   The related statements are additionally supplemented in the Section 4.4.2 (Contributions of different predictors to the forecast skill) at **line 500** as the **blue** text below:

[revised manuscript text omitted]

The related statements are also additionally supplemented in the Section 5 (Discussions and conclusions) at the **line 845** as the **blue** text below:

*"The extensive experiments demonstrate that the ENSO forecast model with a multivariate air-sea coupler (ENSO-ASC) is a powerful tool for analysis of ENSO-related complex mechanisms. Meteorological research does not only pursue skilful models and accurate forecasts, but requires a comprehensive understanding of the potential dynamical mechanisms.* **In the future, we will extend our model to more global physical variables with informative vertical layers, such as the thermocline depth, and the ocean temperature heat content, to explore the global spatial remote teleconnections, temporal lagged correlations, and the optimal precursor etc.***"*

The related references are shown as following and also added into the manuscript:

**Other typos corrections**
When we improve our manuscript, we also find some typos and statement errors, which have been corrected as following:

**Line 525:** "little improvement" → "a little improvement"
**Line 630:** "an 18-month forecasts" → "long-term forecasts"
**Line 655:** "out" → "our"

Thank you again for your positive comments and valuable suggestions to improve the quality of our manuscript.

On behalf of all the co-authors, best regards,
Bo Qin

---

## Author Comment (AC2)

Dear Chief Editor:

Thank you very much for your professional comments concerning our manuscript "ENSO-ASC 1.0.0: ENSO Deep Learning Forecast Model with a Multivariate Air–Sea Coupler" (ID: gmd-2021-213). Those comments are all very valuable and helpful for revising and improving our manuscript, we have updated our manuscript as your suggestion. The point-by-point responses are as following:

**Comment 1:** In your work, it is of the utmost importance that you publish the input sub-datasets used for ENSO-ASC and the output data. Therefore, please, post your data in one of the appropriate repositories.

**Response:** Thank you for spotting our crucial neglects in the datasets used for ENSO-ASC. We have created a repository to store the data we used, including the training/validation dataset and the model output examples, and updated the corresponding statements at **line 862** *(Data availability)* as the **blue** text below:

"*Thanks to NOAA/CIRES, Remote Sensing System, and China Meteorological Administration for providing the historical geoscience data and analysis tools. (https://rda.ucar.edu/, http://www.remss.com/, https://cmdp.ncc-cma.net, last access: 8 Jul 2021).* **The related training/validation datasets can be also accessed at https://doi.org/10.5281/zenodo.5179867**"

**Comment 2:** In the README file of the model, you mention several versions of python or CUDA necessary for your work. This is precisely the kind of information that you must mention in the manuscript and the Code Availability section.

**Response:** Thank you so much for your professional comments. We have updated the corresponding statements at **line 860** *(Code availability)* as the **blue** text below:

"**The source code of the ENSO-ASC is available in the Git repository: https://github.com/BrunoQin/ENSO-ASC (last access: 14 August 2021), which is implemented by Python 3.6 (or 3.7) and CUDA 11.0**. *The present version of ENSO-ASC 1.0.0 is available at https://doi.org/10.5281/zenodo.5081793.*"

**Comment 3:** There is no license listed in the Zenodo repository. For ENSO-ASC (it reads other), but in the uploaded material, there is not a License file. If you do not include a license, the code continues to be your property and can not be used by others. Therefore, when uploading the model's code to Zenodo, you could want to choose a free software/open-source (FLOSS) license. We recommend the GPLv3. You only need to include the file 'https://www.gnu.org/licenses/gpl-3.0.txt' as LICENSE.txt with your code. Also, you can choose other options that Zenodo provides: GPLv2, Apache License, MIT License, etc.

**Response:** We thank the reviewer for this valuable comment. We have updated our model repository and supplement the GPLv3 license into the latest repository as the comment said.

**Comment 4:** In the files of the model, it reads several times "Linux". The correct way of naming it is "GNU/Linux"; Linux is only the kernel of the operative system

**Response:** Thank you for pointing out our mistake very much. We have also replaced the related "Linux" to "GNU/Linux" in the "README.md" file in our model repository.

Thank you again for your positive comments and valuable suggestions to improve the quality of our manuscript.

On behalf of all the co-authors, best regards,
Bo Qin

---

## Author Comment (AC3)

Dear Editor and Reviewer:

Thank you very much for your insightful comments concerning our manuscript "ENSO-ASC 1.0.0: ENSO Deep Learning Forecast Model with a Multivariate Air–Sea Coupler" (ID: gmd-2021-213). Those comments are all very valuable and helpful for revising and improving our manuscript, we have studied comments carefully and have made revisions. The point-by-point responses are as following:

**Comment 1:** In the ablation experiment, "The calculation of this variable contains SST, so the effect of the extra introduction of upper ocean heat content will be weakened" is at L533. I have a suggestion: if using upper ocean heat content to take place the SST in the model, how will the ENSO-ASC perform?

**Response:** Thank you so much for your professional attitude and insightful suggestion. This is indeed a valuable question for investigating the effects of different predictors on an ENSO deep learning forecast model. The upper ocean heat content is a very concerned variable, which can reflect the vertical and horizontal propagations of ocean waves and help interpret the dynamical mechanisms of ENSO. Therefore, as the comment says, we supplement a control experiments to investigate the model performance by replacing SST with upper ocean heat content in the model input.

We conduct the comparison by two modified ENSO-ASCs with the same output of SST + u-wind, v-wind, rain, cloud, and vapor, while with the different input. One is upper ocean heat content + u-wind, v-wind, rain, cloud, and vapor (**EXAM**), the other is SST + u-wind, v-wind, rain, cloud, and vapor (**CTRL**). We find that the forecast skill of **EXAM** is slightly lower than that of **CTRL** (depicted as Table 4). The upper ocean heat content is the average of the oceanic temperature from sea surface to upper 300m, which is crucial to represent the deeper sea temperature beyond sea surface. However, our model is designed to forecast SST. We think that using the upper ocean heat content as a predictor for our model inevitably introduces more noise, which extracts the features of oceanic temperature not only from sea surface but also from deeper ocean. Actually, according to our extensive experiments, we find it is a positive determination that the model should select the physical variable we want to forecast as one of predictors.

We also supplement the related statements from the start of **line 545** as the blue text below:

"Among the three extra added physical variables, the upper ocean heat content is a very concerned variable, which can reflect the vertical and horizontal propagations of ocean waves and help interpret the dynamical mechanisms. Therefore, we conduct the comparison via two modified ENSO-ASCs with the same output of SST + u-wind, v-wind, rain, cloud, and vapor, while with the different input. One uses upper ocean heat content + u-wind, v-wind, rain, cloud, and vapor, marked as EXAM, another uses SST + u-wind, v-wind, rain, cloud, and vapor, marked as CTRL. The results are shown in Table 4.

**Table 1: Model performance comparison when using upper ocean heat content to replace SST in the input**

| Model paradigm | 12-month SSIM / PSNR | 15-month SSIM / PSNR | 18-month SSIM / PSNR |
| --- | --- | --- | --- |

| Model paradigm | | | |
|---|---|---|---|
| **CTRL:** *SST + others* → *SST + others* | 92.65 / 22.05 | 90.31 / 20.97 | 87.53 / 18.17 |
| **EXAM:** *upper ocean heat content + others* → *SST + others* | 90.96 / 20.87 | 88.45 / 18.23 | 84.76 / 14.90 |

Note: Model paradigm represents the input and the output for the ENSO-ASC, where → means "forecast". "Others" is five variables, including u-wind, v-wind, rain, cloud, and vapor. The first row is the control experiment, which is the same with the result in Table 3, and the second row is the examined experiment, which only replaces SST by upper ocean heat content in the model input.

The forecast skill of EXAM is slightly lower than CTRL. The upper ocean heat content is the average of the oceanic temperature from sea surface to upper 300m. When using it as a predictor to forecast SST, our model will extract the features of oceanic temperature not only from sea surface but also from deeper ocean, which inevitably introduces more noise. This may be a reason for the above result. Generally, the model should select the physical variable to be predicted as one of predictors to obtain higher forecast skill."

**Comment 2:** The initial letter of the sentence should be uppercase and some mistakes are found at line 103 and line 222.

**Response:** Thanks for your comment. We have read through the full text and corrected all misspelling and grammatical errors, including **Line 103** and **Line 222**.

**Comment 3:** L374, "N40°-S40°, E160°-W90°", should be expressed as the region of 40°N-40°S, 160°E-90°W.

**Response:** Thanks for your comment. We have corrected the related text in the **Line 374**. In addition, we also modify the statements in the legend of Figure1 as the following blue text:

"Figure 1: Most concerned regions in ENSO events. The blue rectangle covers the Niño3 region ($5^{\circ}$N-$5^{\circ}$S, $150^{\circ}$W-$90^{\circ}$W), and the green rectangle covers the Niño4 region ($5^{\circ}$N-$5^{\circ}$S, $160^{\circ}$E-$150^{\circ}$W)."

**Comment 4:** In Figure 6, the text looks too small.

**Response:** Thank you for your reminding, and it is really a good suggestion to improve the whole quality of our manuscript. We have enlarged the font size and image size of Figure6. In addition, we also check and enlarge the size of other figures in our manuscript to make them more clearly.

Thank you again for your positive comments and valuable suggestions to improve the quality of our manuscript.

On behalf of all the co-authors, best regards,
Bo Qin